

# Combined linear regression and Monte Carlo approach to modelling exposure age depth profiles

Yiran Wang[1, 2], Michael E. Oskin[1]

[1]Department of Earth and Planetary Sciences, UC Davis, Davis, 95616, USA

[2]Earth Observatory of Singapore, Nanyang Technological University, 639798, Singapore

*Correspondence to*: Yiran Wang (yrwwang@ucdavis.edu)

**Abstract.** We introduce a set of methods for analyzing cosmogenic-nuclide depth profiles that formally integrates surface erosion and muogenic production, while retaining the advantages of the linear inversion. For surfaces with erosion, we present solutions for both erosion rate and total eroded thickness, each with their own advantages. For practical applications, erosion

must be constrained from external information, such as soil-profile analysis. By combining linear inversion with Monte Carlo simulation of error propagation, our method jointly assesses uncertainty arising from measurement error and erosion constraints. Using example depth profile data sets from the Beida River, northwest China and Lees Ferry, Arizona, we show that our methods robustly produce comparable ages for surfaces with different erosion rates and inheritance. Through hypothetical examples, we further show that both the erosion rate and eroded-thickness approaches produce reasonable age

estimates so long as the total erosion less than twice the nucleon attenuation length. Overall, lack of precise constraints for erosion rate tends to be the largest contributor of age uncertainty, compared to the error from omitting muogenic production or radioactive decay.

## 1 Introduction

In-situ cosmogenic nuclide (CN) dating, especially with [10]Be, is a widely applied tool to estimate landform ages (e.g., Granger

et al., 2013). These dates are affected by landscape processes that either remove or add CNs, lending uncertainty that may be difficult to assess without additional information. Ages of landforms constructed from sediments, such as a stream terrace, may be affected by CNs acquired by the sediments prior to deposition, termed inheritance, leading to erroneously older dates (Brocard et al., 2003; Hancock et al., 1999; Repka et al., 1997). Conversely, even a low rate of erosion of a landform after its formation will bias surface-exposure ages younger (Lal, 1991). Under the condition of no erosion, the depth-profile approach,

first developed by Anderson et al. (1996), provides a robust technique for estimating surface age and inheritance from a landform comprised of sediments. The effect of erosion, however, is difficult to discern from a depth profile of CN concentrations, leading to a trade-off between model age and erosion rate. Though it is theoretically possible to solve for erosion rate with sufficient number, precision, and depth of sampling (Brocard et al., 2003), realistic sampling scenarios require external constraints of erosion to fully assess a landform age.





There are generally two groups of approaches from which the surface exposure age can be estimated from a CN depth profile. The first group relies on linear inversion of the relationship between concentration and nucleogenic production rate at depth (i.e., Anderson et al., 1996). This approach, as originally formulated, accounts for nucleon (neutron and proton) spallation production of CNs that make up ~98% of surface production and decreases sharply within the upper two meters of sediment. Muons, accounting for the other 2% of surface production, penetrate much deeper than nucleons (Braucher et al., 2003;

Heisinger et al., 2002b, 2002a), such that muogenic production barely decreases within the upper two meters of a depth profile and therefore may be ignored, to first-order (Figure 1a). This inversion approach has the advantage of being straightforward to apply to determine an exposure age without any prior knowledge. However, currently applied linear inversion techniques do not fully account for measurement uncertainty in model ages, and also do not explicitly account for the effects of erosion. In addition, ignoring muogenic production could lead to minor overestimation of surface age and significant overestimation

of inheritance, especially for surfaces undergoing erosion. The second group of approaches uses forward modelling to find best-fit depth-concentration curves, such as with $\chi^2$ minimization (e.g. Braucher et al., 2009; Hidy et al., 2010; Matsushi et al., 2006; Riihimaki et al., 2006), or Bayesian inference (e.g. Laloy et al., 2017; Marrero et al., 2016). These approaches have the advantage of accounting for muogenic production and can include erosion into the inversion process. However, the accuracy and efficiency of these approaches largely rely on researchers' prior knowledge of the surface age and inheritance.

In this paper, we present a combined linear regression and Monte-Carlo approach to analyzing $^{10}$Be depth profiles that formally integrates surface erosion and muogenic production into exposure age modelling. This approach builds upon the simplicity and minimum prior knowledge needed for the linear regression approach, but expands its application to surfaces with independently constrained erosion histories and increases the accuracy of the age and inheritance results by taking muogenic production into account. To demonstrate application of this approach, we examine an example sample site from our previously

published work (Wang et al., 2020), and we re-analyze the Lees Ferry stream terrace site example from Hidy et al. (2010) to compare our method with their Monte-Carlo $\chi^2$ minimization approach. We also discuss the trade-off of surface erosion with exposure age estimation, the impact of muons and radioactive decay on age and inheritance calculations.





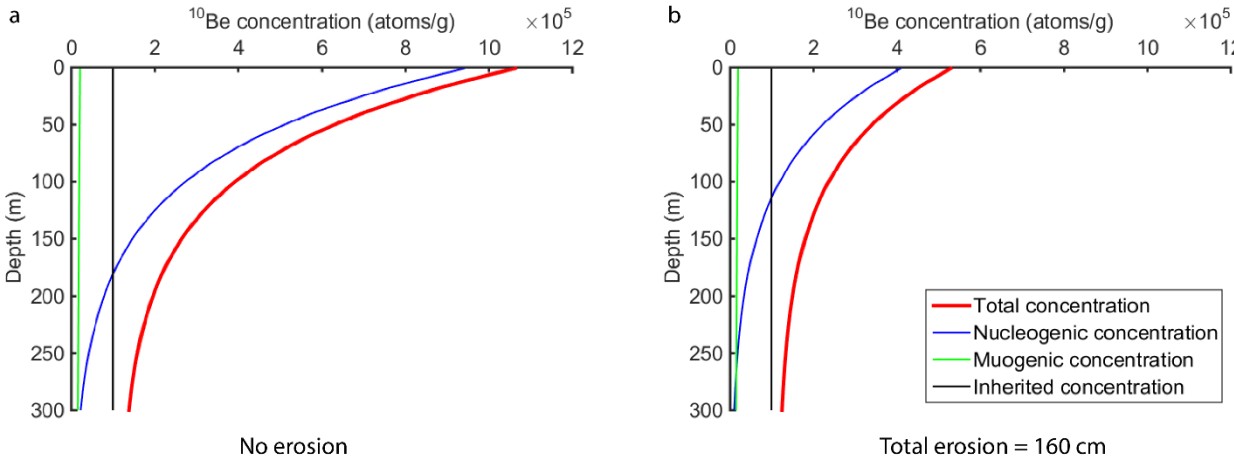

**Figure 1 Depth-concentration profiles with different contributing components for a hypothetical surface. All conditions are the same**
**for both figures: total surface production rate, 10 atoms/(g\*yr); sediment density, 2g/cm3; the relative contributions of nucleons and muons (negative and fast) to the total $^{10}$Be production are 97.85%, 1.5% and 0.65%, with $\Lambda$ equal to 160 g/cm$^2$, 1500 g/cm$^2$ and 5300 g/cm$^2$, respectively ((Braucher et al., 2003)). a. Surface with zero erosion. b. Surface with steady erosion, eroded thickness equal to two attenuation length (160 cm).**

## 2 Methods

### 2.1 General inversion

Under conditions of constant production rate and constant erosion rate, a surface that was exposed at time t would have a concentration of a cosmogenic nuclide ($N_z$) as (Balco et al., 2008; Braucher et al., 2009; Lal, 1991; Lal and Arnold, 1985):

$$N_z(t) = P_{n,0}e^{-\frac{\rho z}{\Lambda_n}}\left(\frac{1-e^{-\left(\frac{\rho r}{\Lambda_n}+\lambda\right)t}}{\frac{\rho r}{\Lambda_n}+\lambda}\right) + P_{m_1,0}e^{-\frac{\rho z}{\Lambda_{m1}}}\left(\frac{1-e^{-\left(\frac{\rho r}{\Lambda_{m1}}+\lambda\right)t}}{\frac{\rho r}{\Lambda_{m1}}+\lambda}\right) + P_{m_2,0}e^{-\frac{\rho z}{\Lambda_{m2}}}\left(\frac{1-e^{-\left(\frac{\rho r}{\Lambda_{m2}}+\lambda\right)t}}{\frac{\rho r}{\Lambda_{m2}}+\lambda}\right) \quad (1)$$

where $P_{n,0}$, $P_{m_1,0}$, and $P_{m_2,0}$ are the surface production rate induced by nucleons, negative muons, and fast muons;
$\Lambda_n, \Lambda_{m1}, and\ \Lambda_{m2}$ are the attenuation lengths of the nucleons and muons (negative and fast), respectively; z is the depth beneath the target surface; $\lambda$ is the decay constant, and r is a constant erosion rate, if applicable. For our purposes, we model ages using $^{10}$Be, with a half-life of 1.39 Myr (Chmeleff et al., 2010; Korschinek et al., 2010; Nishiizumi et al., 2007), due to its wide applicability to quartz-bearing sediments (Cockburn and Summerfield, 2004; Granger et al., 2013; Rixhon et al., 2017). Based on eq. 1, the production of cosmogenic nuclides may be simplified into two major components: the production rate at
specific depth ($P_z$), and the effective exposure age of the site ($T_e$), which is the time that is required to accumulate concentration $N_z$ at production rate $P_z$ without erosion and radioactive decay. Therefore eq. 1 may be rearranged into:

$$N_z(t) = \sum_i P_{zi}T_{ei} \quad (2a)$$

where $P_{zi} = P_{i,0}e^{-\frac{\rho z}{\Lambda_i}}, T_{ei} = \left(\frac{1-e^{-\left(\frac{\rho r}{\Lambda_i}+\lambda\right)t}}{\frac{\rho r}{\Lambda_i}+\lambda}\right), i = n, m_1, m_2 \quad (2b)$





The $^{10}$Be concentration measured from a suite of samples (Figure 1), C, has two components: the in-situ produced concentration, $N_z$ , and the inherited concentration, $C_{inh}$,

$$C = \sum_i P_{zi}T_{ei} + C_{inh}. \qquad (3)$$

Though $^{10}$Be concentration (C) is exponential to the burial depth, based on equation 2 and 3, when there is no surface erosion $(r \approx 0), T_{en} = T_{em1} = T_{em2}$, and therefore eq. 3 can be rearranged as:

$$C = T_e \sum_i P_{zi} + C_{inh}. \qquad (4a)$$

where $T_e = \left(\frac{1-e^{-\lambda t}}{\lambda}\right) \qquad (4b)$

This equation is an update to the linear regression approach first proposed by Anderson et al. (1996) that accounts for both nucleon and muon production, as well as radioactive decay. For the case of no erosion, CN concentration is linear to the sum of production rates via all pathways ($\sum_i P_{zi}$), and $T_e$ and $C_{inh}$ are the slope and intercept of this linear relationship respectively. Therefore, similar to the approach proposed by Anderson et al. (1996), we can apply linear least squares regression to find the slope ($T_e$) and intercept ($C_{inh}$) of the best fit line to the concentration vs. production rate data of the depth profile. The exposure age, factoring in decay, may be calculated directly by rearranging eq. 4b:

$$t = -\frac{\ln(1-T_e\lambda)}{\lambda} \qquad (5)$$

### 2.2 Inversion with erosion rate

For sites with constant erosion rate, *r,* the effective age for each pathway (nucleons or muons) would be different, due to their different attenuation lengths. But an approximation may be made by omitting the muogenic production, on the basis that muogenic production only makes up ~2% of the total surface production (Braucher et al., 2003; Heisinger et al., 2002b, 2002a), and eq. 3 may be further simplified to

$$C = P_{zn}T_{en} + C_{inh}, \qquad (6)$$

Using eq. 6, a linear least squares regression can be applied to find the best-fit $T_{en}$ and $C_{inh}$, which leads to the estimated exposure age

$$t = -\frac{\ln(1-T_{en}B)}{B} \quad (7a) \qquad \text{where } B = \frac{\rho r}{\Lambda_n} + \lambda \qquad (7b)$$

This solution illustrates the utility of separating the age model for finding $T_{en}$ from the effect of erosion rate, contained within the parameter *B.* Considering only nucleons, there is no information from a depth profile of CN concentrations that constrains erosion rate, except for the upper limit of this rate that yields an infinite exposure age when B = 1/$T_{en}$.

### 2.3 Inversion with eroded thickness

For many practical cases, it may be more straightforward to estimate total eroded thickness (D) from field evidence such as through soil-profile analysis, rather than an erosion rate. With eroded thickness, the effective age of each pathway may be rewritten as



$$T_{ei} = \left( \frac{1 - e^{-\left(\frac{\rho D}{\Lambda_i} + \lambda t\right)}}{\frac{\rho D}{\Lambda_i t} + \lambda} \right), i = n, m_1, m_2 \qquad (8)$$

Here we explore the application of this equation with the inclusion of muogenic production. Using a series expansion, we rewrite the effective age related to muons, $T_{em}$, into a fraction, *g,* of the effective age related to nucleons, $T_{en}$. The fraction *g* can be approximated solely from knowledge of the eroded thickness, *D* (see Appendix for derivation):

$$g_i = \frac{T_{emi}}{T_{en}} \approx e^{-\frac{1}{2}\left(\frac{\rho D}{\Lambda_{mi}} - \frac{\rho D}{\Lambda_n}\right) + \frac{1}{24}\left[\left(\frac{\rho D}{\Lambda_{mi}}\right)^2 - \left(\frac{\rho D}{\Lambda_n}\right)^2\right]}, i = 1, 2 \qquad (9)$$

Bringing $g_i$ into eq. 3, we have

$$C(z) = P_{zn}T_{en} + P_{zm_1}g_1 T_{en} + P_{zm_2}g_2 T_{en} + C_{inh} = P_{ze}T_{en} + C_{inh},$$

$$P_{ze} = \left(P_{zn} + P_{zm_1}g_1 + P_{zm_2}g_2\right) \qquad (10)$$

where $P_{ze}$ is the effective production rate from both nucleons and muons under the condition of a finite amount of erosion over the lifetime of the deposit. Note that the robustness of the muogenic production approximation (see Appendix) illustrates how erosion depth (or rate) may not be well constrained from concentration-depth profiles alone, even when including muogenic

production, and even though a unique solution for age, inheritance, and erosion rate formally exists (Broccard et al., 2003). Using equation 10, $T_{en}$ and $C_{inh}$ can be found by applying least squares linear regression with known production rates, eroded thickness, and sample concentrations, similar to the general inversion case for no erosion described by equation 4.

To estimate the exposure age, we need to find the solution for

$$f(t) = \left( \frac{1 - e^{-\left(\frac{\rho D}{t\Lambda_n} + \lambda\right)t}}{\frac{\rho D}{t\Lambda_n} + \lambda} \right) - T_{en} = 0 \quad (11)$$

While the complicated form of eq. 11 prohibits a direct solution, t may be found iteratively by applying the Newton's method. Using the derivative of eq. 11,

$$f'(t) = -\lambda e^{-\left(\frac{\rho D}{\Lambda_n}\right)D - \lambda t} - \frac{\rho D T_{en}}{\Lambda_n t^2}, \quad (12)$$

the exposure age can then be iterated from

$$t_{n+1} = t_n - \frac{f(t_n)}{f'(t_n)} \qquad (13)$$

with initial guess, $t_0 = T_{en}$.

## 2.4 Uncertainty treatment with Monte Carlo simulation

In our model (https://github.com/YiranWangYR/10BeLeastSquares), we consider the uncertainty of the exposure age propagated from four different sources: analytical uncertainties of the [10]Be concentration measurements, uncertainty of sample depths, uncertainty of the erosion depth or rate, and the uncertainties related to CN production and decay (i.e., the attenuation

length, production rates, etc.). These uncertainties propagate sequentially, first from [10]Be concentration and sample depths





through least-squares regression process, and second from erosion rate or depth through converting exposure age from the effective exposure age ($T_e$). The uncertainties related to $^{10}$Be production and decay affect both steps.

Because of the limited sample sizes typical of most studies, and the variance in both concentration and depth, we propose a Monte Carlo simulation approach to determine the range of exposure age and inheritance. For each iteration, we randomly

select a group of values (C, z, $P_z$, and r or D, etc.) from their corresponding probability density functions. The slope and intercept ($T_e$ and $C_{inh}$) are found via least-squares linear regression of the concentration versus production rate as a function of depth. Then the exposure age, t, may be calculated using eq. 5, eq. 8, or eq. 11 through 13. Repeating these steps yields a distribution representing the probability of t and $C_{inh}$ of the samples. By increasing the number of iterations, the shape of the resulting probability distribution becomes apparent and the accuracy increases.

## 140  3 Applications

In this section, we apply our model to two published $^{10}$Be depth profile sample sites. One from our own published research of the Beida River T2 terrace of the North Qilian Shan, China (Wang et al., 2020). We use this site to demonstrate the modeling steps in detail. The second site is the Lees Ferry site, an example from Hidy et al. (2010). We use this second site to compare our modeling results with their widely used $X^2$ minimization technique.

### 145  3.1 Beida River Example

#### 3.1.1 Sample site

In western China, the Qilian Shan orogen serves as the northeastern margin and youngest growing portion of the Tibetan plateau. The Beida River is the largest river that flows northward across the western portion of the North Qilian Shan. At least three principal generations of fill terraces (T1, T2, and T3) are preserved along the Beida River inside the Qilian Shan mountain

range. Mapping and dating of these terraces make it possible to understand the aggradation-incision process of the river and interpret the tectonic deformation of the North Qilian Shan (Wang et al., 2020). Our sample site is located on a T2 terrace tread, ~5 km upstream of the mountain front, and more than 200 m above present riverbed. The T2 terrace at this location has been dissected by gullies into several isolated lobes, suggesting that remnant terrace treads might have experienced some degree of surface erosion. Loess of ~130 cm thickness is deposited atop of the terrace tread. A OSL sample at the loess bottom

suggests loess deposition started around 8.3±1.2 kyr (Wang et al., 2020).

We excavated a sample pit ~2 m deep on this T2 terrace tread. The soil profile developed on the terrace fill shows a 20 cm-thick remnant reddened B horizon with clay directly in contact with the base of the loess cover (Supplement figures of Wang et al., 2020). An unknown thickness of the B horizon, along with the original soil A horizon, are missing, and were presumably eroded prior to loess deposition. We therefore interpret that there may have been 20-60 cm of erosion of the terrace tread,

before the onset of loess accumulation. We collected six samples of medium to coarse sand from up to 2 m below the base of the loess. These samples were processed at Arizona State University following standard chemical cleaning and etching





procedures. AMS measurements of these samples were conducted at the Prime Lab of Purdue University. The $^{10}$Be concentrations reported by the Prime Lab are listed in table 1.

**Table 1 Sample information and $^{10}$Be concentration of Beida River T2 site (Wang et al., 2020)**

| Sample ID | Coordinates and elevation | Depth (cm, below loess bottom) | Thickness (cm) | Dissolved quartz (g) | Carrier Mass(g) | Corrected $^{10}$Be/$^9$Be | $^{10}$Be (atoms/g) | 1σ Error (atoms/g) |
|---|---|---|---|---|---|---|---|---|
| BT2-2-20 | | 40 | 10 | 22.21001 | 0.33669 | 1.42E-12 | 1497204 | 22001 |
| BT2-2-45 | | 65 | 10 | 14.73835 | 0.34244 | 6.42E-13 | 1035575 | 24187 |
| BT2-2-75 | 39.5743 N, 97.998 E; | 75 | 10 | 14.79391 | 0.34183 | 3.76E-13 | 602740 | 15467 |
| BT2-2-110 | 2300 m | 130 | 10 | 9.02134 | 0.34213 | 1.64E-13 | 430335 | 16094 |
| BT2-2-150 | | 160 | 10 | 23.50438 | 0.33589 | 3.13E-13 | 310141 | 7535 |
| BT2-2-180 | | 200 | 10 | 30.15846 | 0.30283 | 3.89E-13 | 271258 | 5698 |


**Table 2 $^{10}$Be concentration prior to (C1) and post (C2) loess accumulation, and the production rate at each sample depth.**

| Sample ID | Loess cover (cm) | $C_2$ ($10^4$ atoms/g) | $C_1$ ($10^5$ atoms/g) | $P_z$ ($P_0 e^{-\frac{\rho z}{\Lambda}}$; atom*g$^{-1}$*yr$^{-1}$) |
|---|---|---|---|---|
| BT2-2-20 | | 7.11 ± 1.58 | 14.33 ± 0.39 | 13.82 ± 0.91 |
| BT2-2-45 | | 5.11 ± 1.14 | 9.84 ± 0.36 | 9.94 ± 0.65 |
| BT2-2-75 | | 3.44 ± 0.77 | 5.68 ± 0.23 | 6.69 ± 0.44 |
| BT2-2-110 | 130 | 2.17 ± 0.48 | 4.09 ± 0.21 | 4.22 ± 0.28 |
| BT2-2-150 | | 1.46 ± 0.33 | 2.96 ± 0.11 | 2.84 ± 0.19 |
| BT2-2-180 | | 0.86 ± 0.19 | 2.63 ± 0.08 | 1.68 ± 0.11 |

**Table 3 Values for parameters used in exposure age calculation.**

| Parameter | Values (Wang et al., 2020) | Values (Hidy et a., 2010) |
|---|---|---|
| Surface production rate (nucleon-negative muon-fast muon) (atom*g$^{-1}$*yr$^{-1}$) | 23.4, 0.259, 0.155 | 9.51, 0.145, 0.115 |
| Density (g/cm3) | 2.2 | 2.2-2.5 (uniform distribution) |
| Attenuation (nucleon- negative muon-fast muon) (g/cm2) | 167, 1500, 5300 | 160±5, 1500, 5300 |





| Eroded thickness (cm) | 40±10 (normal distribution) | 0-30 (uniform distribution) |
|---|---|---|
| Erosion rate (cm/kyr) | 0.3±0.05 (normal distribution) | 0-0.32 (uniform distribution) |

### 3.1.2 Exposure age estimation

Because the terrace tread is covered by loess, we need to first estimate the $^{10}$Be concentration at the time of the onset of loess accumulation. Follow the approach introduced by Hetzel et al., (2002), the $^{10}$Be concentration prior to ($C_1$) and post ($C_2$) loess accumulation are calculated and listed in table 2. Parameters we use for inversion are listed in table 3.

Using a normally distributed erosion rate of 0.3±0.05 cm/kyr, we first find the effective exposure age ($T_{en}$) and inheritance (at the time of loess accumulation; $C_{inh}$) by linear regression using eq. 6. The best fit line(s) of the data ($C_1$ and $P_{zn}$) are shown on figure 2a, the fitted depth profile curves are shown on figure 2b. Because our sample site contains very low inheritance (Figure 2d), some inversion results yield non-physical predictions with negative inheritance. These negative inheritance predictions are necessary to estimate the full distribution of the exposure age, but we exclude these from the final inheritance results. The predicted 95% confidence range of $T_{en}$ and $C_{inh}$ after 100,000 iterations ranges from 86.8 to 111.7 kyr and from -3.4 x 10$^4$ to +7.12 x 10$^4$ atoms/g, respectively. Substituting $T_{en}$ and $C_{inh}$ into eq. 7, the range of the exposure age is 107.6-160.8 kyr (95% confidence) prior to loess accumulation (Figure 2e). The possible range of inheritance is 0-7.12 x 10$^4$ atoms/g after excluding negative results. The corresponding eroded thickness is 23-59 cm (Figure 2g).

For the eroded-thickness approach, we also assume a normal distribution for the total erosion, and choose 40 cm as the mean and 10 cm as the standard deviation of the eroded thickness. By applying least squares linear inversion with eq. 10, the best fit line(s) of the data ($C_1$ and $P_{ze}$) are shown on figure 3a, the fitted depth profile curves are shown on figure 3b. Including the muogenic production pathways into calculation leads to a slightly younger (1% shift of the mean) $T_{en}$ value of 85.9-110.7 kyr and a lower inheritance of -9.1 x 10$^4$ to +2.9 x 10$^4$ atoms/g (ranges correspond to the 95% confidence distributions for each value, figure 3c 3d). The corresponding exposure age, calculated following eq. 11-13, is 108.3-154.2 kyr (2σ) prior to loess accumulation (Figure 3e). Excluding the negative results, the possible range of inheritance is 0-2.9 x 10$^4$ atoms/g. The corresponding erosion rate is 0.18-0.42 cm/kyr (Figure 3f and 3g).






**Figure 2 Linear regression results for Beida River T2 terrace data set using the erosion-rate approach after 100,000 iterations. a. Relationship of sample concentration to production rate at depth. Grey lines are the best fit lines through this data set. b. distribution**
**of depth profile models with best fit curves (grey lines). c. Distribution of the effective exposure age ($T_e$); d. Inherited $^{10}$Be concentration prior to loess accumulation. e. Exposure age estimated based on preset erosion rates with $T_e$ value derived from linear regression (Figure 2c). f. Distribution of sampled erosion rates; g. Distribution of total eroded thicknesses predicted by the model. Red lines indicate 2σ confidence error range, green line indicates the median of the distribution, blue line indicates the mean of the distribution.**







**Figure 3 Linear regression results for Beida River T2 terrace using eroded-thickness approach after 100,000 iterations. a. Relationship of sample concentration to production rate at depth. Grey lines are the best fit lines through this data set b. distribution of depth profile models with best fit curves (grey lines). c. Distributions of the effective exposure age ($T_e$); d. Inherited $^{10}$Be concentration prior to loess accumulation. e. Exposure age estimates based on preset erosion rates with known $T_e$ value from linear**
**regression (Figure 3c). f. Distribution of erosion rates predicted by the model; g. Distribution of sampled total eroded thicknesses. Red lines indicate 95% confidence range, green line indicates the median of the distribution, blue line indicates the mean of the distribution.**





### 3.2 Lees Ferry Example

#### 3.2.1 Sample site

This [10]Be depth profile data set was originally reported by Hidy et al. (2010). The sample pit was excavated on top of the M4 (main stem) Colorado River fill terrace at Lees Ferry, Arizona. Based on the soil profile, a total erosion of 0-30 cm is estimated for the sample site. One surface sample and two groups of depth profile samples (a sand profile and a pebble profile) were collected from the pit, but they rejected the results of the pebble profile data, for their poor fit to the depth profile and the estimated age result deviates largely from their independent OSL age constraint. For this site Hidy et al. (2010) applied their

model to estimate an exposure age and inheritance of $83.9^{+19.1}_{-14.1}$ kyr, and $9.49^{+1.21}_{-2.52} \times 10^4$ atoms g$^{-1}$, respectively (95% confidence). The erosion rate of the site was estimated as 0-0.32 cm/kyr. See Hidy et al. (2010) for more details of the sample site, sampling and processing, age results interpretation.

**Table 4 Sample information and [10]Be concentration of Lees Ferry sample site (Hidy et al., 2010)**

| Sample ID | Coordinates and elevation | Depth (cm) | Thickness (cm) | Dissolved Quartz (g) | Carrier Mass (g) | Corrected [10]Be/[9]Be | [10]Be Concentration (atoms/g) | 1s Total Measured Error | $P_z$ (atom*g$^{-1}$*yr$^{-1}$) |
|---|---|---|---|---|---|---|---|---|---|
| GC-04-LF-404.30s | | 27.5 | 5 | 45.2566 | 0.308 | 1.2769E−12 | 568744 | 17347 | 6.35 ± 0.48 |
| GC-04-LF-404.60s | | 57.5 | 5 | 45.9469 | 0.3 | 9.5176E−13 | 406713 | 11469 | 4.09 ± 0.48 |
| GC-04-LF-404.100s | 36.853°N, −111.606°W; 985 m | 97.5 | 5 | 50.1042 | 0.3123 | 7.1640E−13 | 292243 | 8972 | 2.27 ± 0.39 |
| GC-04-LF-404.140s | | 137.5 | 5 | 51.1421 | 0.3034 | 5.2302E−13 | 203072 | 6234 | 1.26 ± 0.29 |
| GC-04-LF-404.180s | | 177.5 | 5 | 55.3693 | 0.3085 | 4.3112E−13 | 157209 | 4921 | 0.7 ± 0.2 |
| GC-04-LF-404.220s | | 217.5 | 5 | 55.1112 | 0.2974 | 3.7997E−13 | 134198 | 3892 | 0.39 ± 0.13 |

**3.2.2 Exposure age estimation**

As in the original study, we apply our modelling approaches to the sand depth-profile data (Table 4). In order to compare with results reported by Hidy et al. (2010), we use the same values as they did for all parameters wherever possible (Table 3). Similar to the Beida River profile, we estimate the exposure age with both erosion-rate and eroded-thickness approaches. With the erosion-rate approach, we use a uniformly distributed erosion rate of 0-0.32 cm/kyr (Figure 4f). We invert the

effective exposure age ($T_{en}$) and inherited concentration ($C_{inh}$) based on [10]Be production rate and concentration at each sample depth. The best fit lines and curves are in figure 4a and 4b. The estimated range of $T_{en}$ and $C_{inh}$ values are 66.1-79.6 kyr and 9.45-12.82 x 10$^4$ atoms/g, respectively (95% confidence, Figure 4c, 4d). The estimated exposure age is between 70.5-96.4 kyr (Figure 4e).





With the eroded-thickness approach, we use a uniformly distributed 0-30 cm thickness (Figure 5g). We invert the effective
exposure age ($T_e$) and inherited concentration ($C_{inh}$) based on effective $^{10}$Be production rate ($P_{ze}$) (eq. 10) and $^{10}$Be
concentration at each sample depth. The best fit lines and curves are in figure 5a and 5b. The estimated range of $T_e$ and $C_{inh}$
values are 65.7-79.0 kyr and 7.78-11.11 x $10^4$ atoms/g, respectively (95% confidence; Figure 5c and 5d). The estimated
exposure age is 70.6-95.1 kyr (95% confidence; Figure 5e).

**Figure 4 Linear regression results for Lees Ferry data set with erosion-rate approach after 100,000 iterations. a. Relationship of
sample concentration to production rate at depth. Grey lines are the best fit lines for the data. b. distribution of depth profile models**




with best fit curves (grey lines). c. Distribution of effective exposure age ($T_e$); d. Distribution of inherited $^{10}$Be concentration. e. Exposure age estimated based on preset erosion rates with $T_e$ values derived from linear regression (Figure 4c). f. Distribution of sampled erosion rates. g. Distribution of total eroded thicknesses predicted by the model. Red lines indicate 95% confidence range, green line indicates the median of the distribution, blue line indicates the mean of the distribution.


**Figure 5 Linear regression results for Lees Ferry data set with eroded-thickness approach after 100,000 iterations. a. Relationship of sample concentration to production rate at depth. Grey lines are the best fit lines for the data. b. distribution of depth profile**





**models with best fit curves (grey lines). c. Distribution of the effective exposure age ($T_e$); d. Distribution of inherited $^{10}$Be concentration. e. Distribution of exposure age estimated based on preset erosion rates with $T_e$ values derived from linear regression (Figure 5c). f. Distribution of erosion rates predicted by the model. g. Distribution of total eroded thicknesses. Red lines indicate 95% confidence error range, green line indicates the median of the distribution, blue line indicates the mean of the distribution.**

## 4 Discussion

### 4.1 Modeled age results

For Beida River T2 terrace, the age prior to loess accumulation estimated with erosion rate is 107.6-160.8 kyr, and the age estimated with eroded thickness is 108.3-154.2 kyr. Compare to the age we reported previously (107.9-164.5 kyr; Wang et al., 2020), the age estimated here (erosion-rate approach) is slightly younger (1.5% shift of the mean age). This shift occurs because the depth distributions for each sample were not sampled independently in the original paper. For the age estimated with the eroded-thickness approach, the mean is 4% younger than in the 2020 paper, while the 95% error range is 19% smaller. These

differences come from three different sources. First, a 1.5% shift from independently sampling depth for each measurement. Second, taking muogenic production into account leads to slightly younger age estimations and lower inheritance estimations – an issue we explore further below. Third, the corresponding erosion rate distributions are slightly different for the two approaches (Figure 2f, 3f). Sources 2 and 3 combined leads to 2.5% shift of the mean age and to 15% narrowing of the error range.

It is important to note that, for sites with low inheritance like the Beida River T2 site, permitting negative inheritance results in the resulting distribution is essential to accurately estimating the best-fit exposure age. Truncating the exposure-age distribution by removing negative inheritance results will bias the best-fit age younger because the underlying inheritance distribution will be biased higher. For example, the true age of a surface with zero inheritance would lie to the extreme older tail of such a truncated age distribution, and would thus be excluded at 95% confidence. If instead, negative inheritance results

were not discarded, the age true surface age would lie at the expected (mean) value of the full, untruncated distribution. For our realistic example, if we exclude negative inheritance from our age inversion for T2, the resulting exposure age distribution (pre-loess accumulation) would span 101.6-135.1 kyr at 95% confidence, excluding almost 20 kyr from the older tail. The best-fit age value declines by ~10% in the truncated distribution. We further note that this issue related to low inheritance samples not only affects our least squares inversions, but also affects other exposure-age estimation approaches.

For the Lee's Ferry site, our age estimations of 71.0-96.4 kyr (erosion-rate approach) and 70.5-95.0 kyr (eroded-thickness approach) are very similar to each other, with the erosion-rate age slightly older due to the exclusion of muogenic production. These estimates generally agree with Hidy et al. (2010)'s result, 69.8-103 kyr, but the uncertainty at 95% confidence is smaller with our inversion-based Monte-Carlo approach. We suggest this arises from differences between the two approaches. Specifically, though both methods attempt to minimize sum of the squares of the residuals, the least squares linear inversion

samples C, r, and z, and the inversion only finds one set of best-fit t and Cinh from each sample. Conversely, the forward-model $\chi^2$-minimiziation approach employed by Hidy et al. (2010) randomly samples t, and Cinh, in addition to r and z values





from proposed ranges then calculates the $\chi^2$ values to find the sets of results that fall within 95% confidence of measured concentration data. The inheritance estimated using our erosion-rate approach is 9.45-12.82 x $10^4$ atoms/g, which is significantly larger than Hidy et al. (2010)'s result, 6.97-10.7 x $10^4$ atoms/g. This is mainly because the erosion-rate inversion

(eq. 6) does not account for muogenic $^{10}$Be. On the other hand, the inheritance estimated using our eroded-thickness approach is 7.78-11.11 x $10^4$ atoms/g, which is only slightly larger than Hidy's result. We attribute this difference to the slightly narrower range of best-fitting exposure age estimates found using our inversion approach.

## 4.2 Sources of Error

### 4.2.1 Surface erosion

Surface erosion and its uncertainty constitute a major source of error in exposure age estimation. With the same surface $^{10}$Be concentration, higher erosion rate and/or larger eroded thickness would result in an older effective surface age (e.g. Figure 1). If a surface is sufficiently old, or if the erosion rate is sufficiently high, the CN build up at surface will reach equilibrium with nuclides removed through erosion (Lal, 1991). Figure 6a shows the relationship between erosion rates and surface ages. This figure suggests that once the eroded thickness exceeds the mean attenuation length of nucleon spallation ($\frac{\Lambda n}{\rho}$), the slope of the

age versus erosion rate relationship decreases so as to make the age determination poor. Once the eroded thickness exceeds twice of the attenuation length of spallation, the age versus erosion rate curve flattens so much that it becomes effectively impossible to estimate surface age. On the other hand, the age-eroded thickness curve does not flatten as much (Figure 6b), and therefore it is theoretically possible to use the eroded thickness to determine surface age even when total erosion exceeds twice of the attenuation length. In practice, however, surfaces with a large amount of erosion would subject to large

uncertainties and the erosion history may be too complex for the constant erosion rate assumption to be valid, casting doubt on the utility of $^{10}$Be exposure dating for such cases.

Erosion affects the uncertainty of exposure age estimation in two different ways. First, the uncertainty on the final age gets larger as the erosion rate or thickness increases because of the non-linear relationship between age and erosion rate or thickness (Figure 6). Second, the age uncertainty will increase further through propagation of the uncertainty of the erosion rate. This

suggests that when excavating depth profile pits in surfaces subject to erosion, it is crucial to document surface texture and analyze the soil profiles to estimate eroded thickness, for a small deviation from the true erosion rate or depth would lead to a large bias in the resulting exposure age.



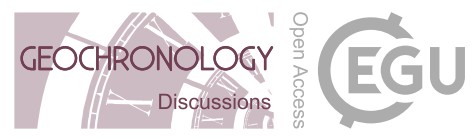

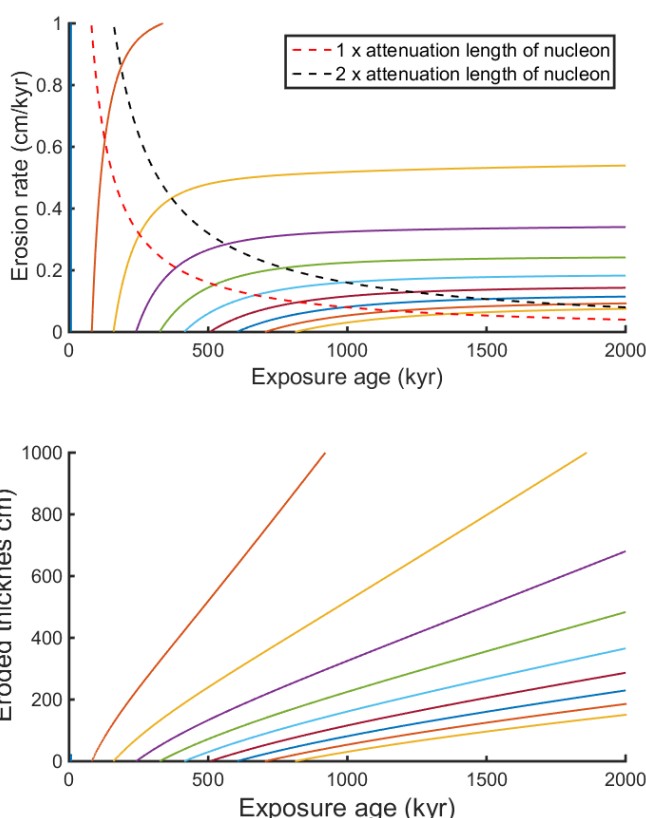

**Figure 6 a. The relationship between erosion rate and exposure age. Each colored line representing the age-erosion relationship of a specific depth profile (or surface concentration). b. The relationship between eroded thickness and exposure age; the color coding is similar to fig. 6a. The parameters used for this simulation are: a total production rate of 15 atoms/g, a density of 2 g/cm³; the relative contributions of nucleons and muons (negative and fast) to the total $^{10}$Be production were 97.85%, 1.5% and 0.65%, the relative attenuation lengths are 160 g/cm², 1500 g/cm² and 5300 g/cm².**

### 4.2.2 Radioactive decay

For CNs such as $^{10}$Be, the actual exposure age, t, is always larger than the effective exposure age, $T_e$, due to radioactive decay (eq. 5). The error resulting from ignoring decay grows larger as the surface age increases (Figure 7). For young surfaces (<200 ka) with zero erosion, excluding radioactive decay underestimates the age by less than 5 %. For older surfaces, i.e., a surface with an age of 1 Myr, ignoring radioactive decay would result in ~30 % of underestimation. All of the approaches developed in this paper take decay into account.






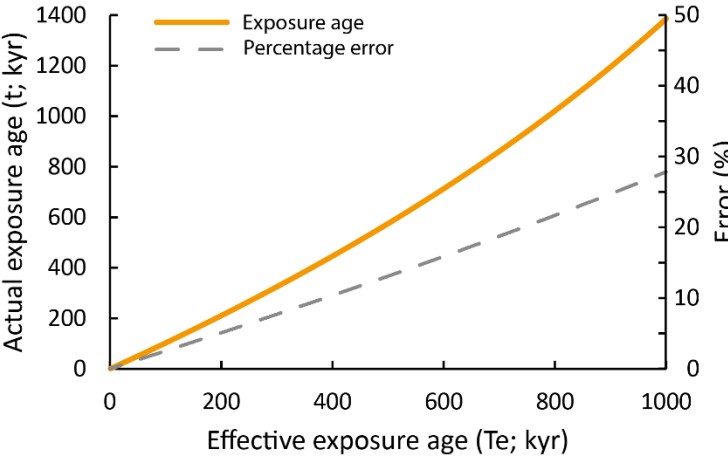

**Figure 7 The relationship between actual and estimated exposure age when radioactive decay is ignored.**

### 4.2.3 Muogenic production

Muogenic production affects the accuracy of the estimated surface ages differently for the various approaches considered here. For surfaces with no erosion, muons may be fully incorporated into the inversion (eq. 4), therefore the uncertainty only comes

from the uncertainties of parameters related to muogenic production (attenuation length and production rate). Ignoring muons and only relying on the relationship between [10]Be concentration and the nucleon spallation production rates (Anderson et al., 1996) leads to slight overestimation of exposure age and inheritance. A demonstration of this effect is shown on figure 8a. By ignoring muons (treating the total concentration as the result of nucleon spallation and inheritance), the line fitting the $P_{zn}$ vs. C data shifts upward and becomes slightly (and hardly recognizably) steeper (Figure 8a). This is because the inversion process

attributes a small portion of the muogenic concentration to nucleon spallation, and a larger portion is attributed to inheritance. When surface erosion is present, muogenic production plays a larger role determining the age (Figure 1b). In fact, with the erosion-rate approach, the error due to ignoring muons grows exponentially with erosion rate (Figure 8b).

To further investigate how erosion affects age and inheritance estimates, with or without inclusion of muogenic production, we compute exposure ages under five different erosional conditions: total eroded thickness equivalent to 0.5, 1, 2, 3, and 5

times the mean attenuation length for spallation (Figure 9). It is worth noting that 3 and 5 times the mean attenuation length for spallation are included as a theoretical analysis; we do not recommend exposure dating for surfaces with such large eroded thicknesses. With the erosion-rate approach, which omits muogenic production, the error introduced is relatively small (less than 2% overestimate) when the total erosion is under one attenuation length for spallation. The age error increases to just under 5% when the total erosion increases to two times the attenuation length (Figure 9a). Above this amount of erosion, the

error grows drastically until no meaningful result can be found with this approach. The eroded-thickness approach, which includes an approximate solution for the muogenic contribution (Appendix A), reduces the error considerably (Figure 9b).



Compared to the erosion-rate approach, the eroded-thickness approach provides very accurate age estimations even with a large amount of erosion.

To assess errors for inheritance, we factor out production rate and express inheritance error in terms of years. Like exposure

age, the error in estimated inheritance is related to surface erosion and exposure age, but the error range is not proportional to mean inheritance. As demonstrated in figure 9c and 9b, the amount of error is one order of magnitude smaller for the eroded-thickness approach compared to the erosion-rate approach. It is worth noting that the errors discussed herein are calculated based on an idealized depth profile, such that all the sample concentrations perfectly fit the expectation. In practice, natural variation of measurements about the expectation leads to larger errors than this idealized case.

Reassessing our age estimates for the Beida River T2 terrace, the ~2.5% difference between the means of the exposure age estimated with muon-included (eroded thickness) and muon-omitted (erosion rate) approach is slightly larger than our idealized model predicts (~1%). For the Lees Ferry site, we find less than 1% difference between the ages estimated with or without muon contributions, matching the expectations from our hypothetical example (~0.5%).

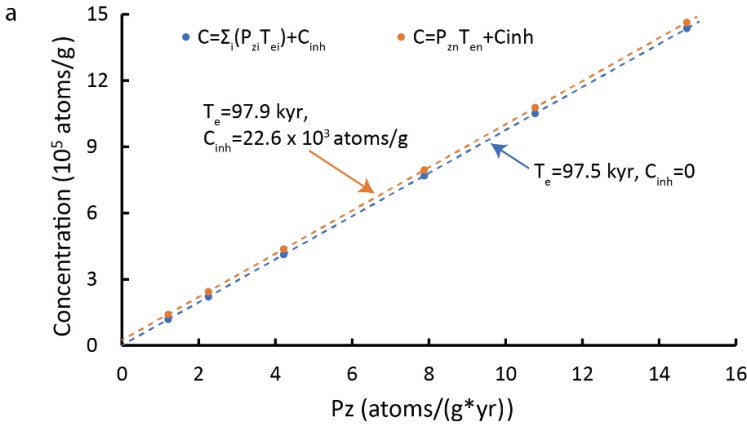

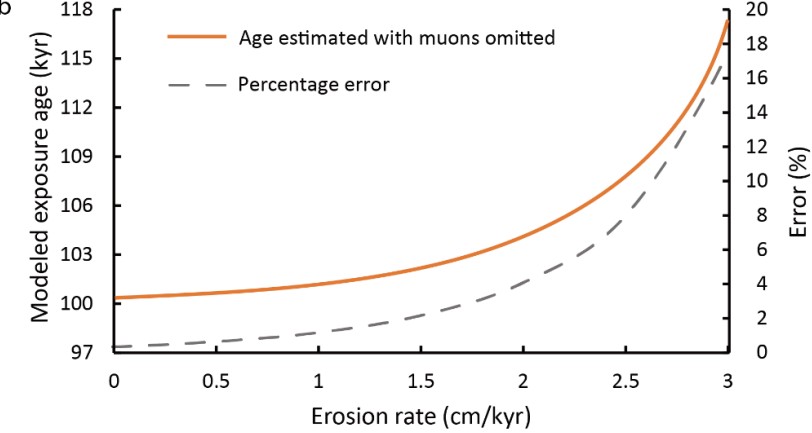

**Figure 8 Case study illustrating the solutions for a surface exposed for 100 kyr; a. best fit lines for the surface with zero erosion rate, data same as table S1. b. Modeled exposure ages (left) and model error (right) versus erosion rate. The parameters used for this simulation are: a total production rate of 15 atoms/g, a density of 2 g/cm³; the relative contributions of nucleons and muons (negative**





and fast) to the total $^{10}$Be production were 97.85%, 1.5% and 0.65%, the relative attenuation lengths were 160 g/cm$^2$, 1500 g/cm$^2$ and 5300 g/cm$^2$.

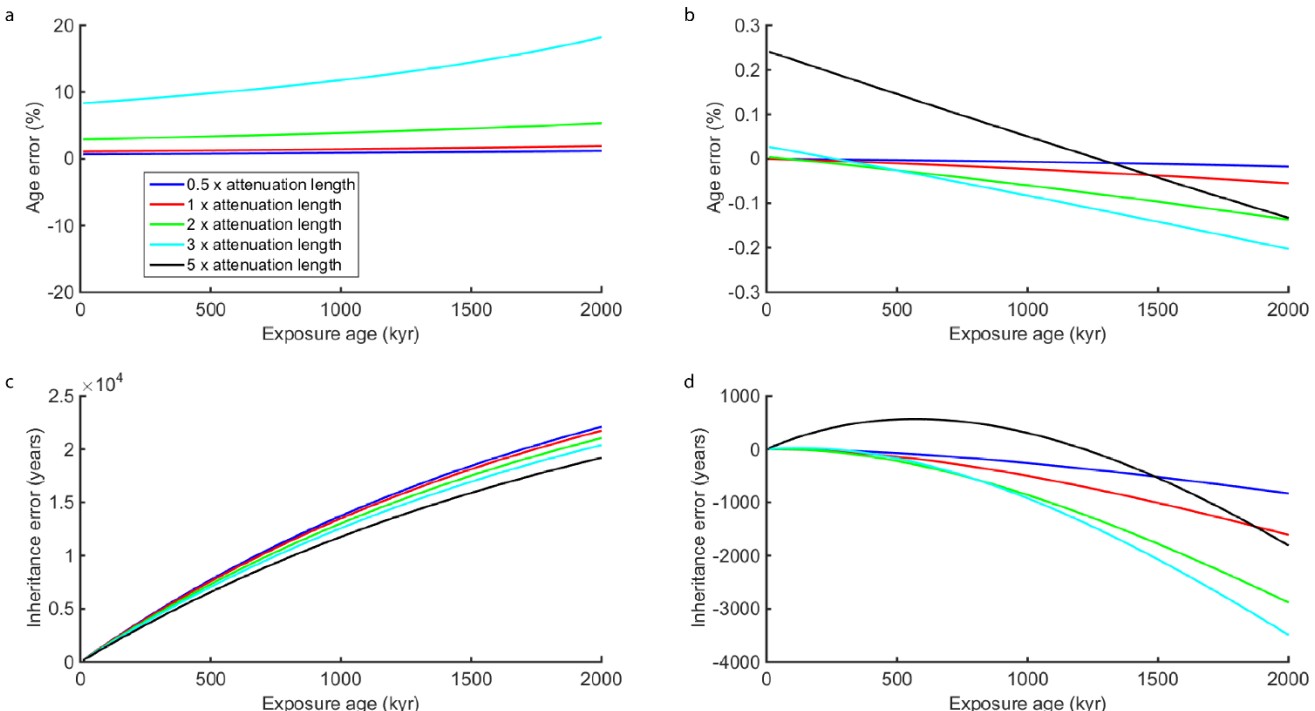


**Figure 9 Degrees of errors vs. exposure age show the advantage of the eroded-thickness approach with approximate solution for muogenic production (b and d) over the erosion-rate approach without muogenic production (a and c). Each line represents a modeled surface that has undergone various exposure times, but with the same total eroded thickness, expressed as a multiple of the attenuation length, $\dfrac{\rho}{\Lambda_n}$. a. Age error using erosion-rate approach (eqs. 6-7), when total erosion reaches 5 times attenuation length, no**

**meaningful result can be found; b. Age error using eroded-thickness approach (eqs. 8-13); c. Inheritance error (expressed in concentration divided by surface production rate) using erosion-rate approach; d. Inheritance error using eroded-thickness approach. The relative contributions of nucleons and muons (negative and fast) to the total $^{10}$Be production are 97.85%, 1.5% and 0.65%, with $\Lambda$ equal to 160 g/cm$^2$, 1500 g/cm$^2$ and 5300 g/cm$^2$, respectively (Braucher et al., 2003).**

**4.2.4 Trade-offs between error sources and age estimation**

In previous sections, we discussed how erosion rates, radioactive decay, and muogenic production each individually affect exposure age estimates and error. In general, the muogenic $^{10}$Be production contributes least to the uncertainties related to surface exposure dating, compared to the effects of neglecting decay and, especially, the impact of erosion. We show here that radioactive decay may be easily corrected for after finding the effective age with linear regression (eq. 5, 7, 11-13). Surface erosion, and its uncertainty, thus generally constitutes the largest source of uncertainty for surface-exposure dating.

With the erosion-rate approach, the degree of error depends on the surface age, and more importantly, the total amount of surface erosion. This method is simple and accurate enough for most exposure age applications (Figure 6a, 9a). When total



erosion exceeds one to two attenuation lengths, however, the erosion-rate approach is either unable to produce meaningful results, or the uncertainty would be too large (Figure 6a). The eroded-thickness approach is slightly more complex, but also more accurate than the erosion-rate approach, because it incorporates the muogenic contribution. Our approximation of the muogenic contribution (eq. 9; figure 9b) is very robust even with a large total erosion depth. Consequentially, the inheritance estimation is also more accurate with the eroded-thickness approach (Figure 9c, 9d).

As we show on fig. 6, a small change of erosion rate or eroded thickness may lead to large differences in surface age estimation, especially for the erosion-rate approach. Therefore, we suggest when dating surfaces with erosion, careful examination of the soil profile or other independent evidence of surface preservation is necessary to provide the best constraint possible on the erosion depth. In addition, it is important to consider the assumption of a constant erosion rate at the sample site. It is entirely possible that a sample site may have experienced episodic erosional episodes instead of constant erosion rate, which would lead errors in the age not accounted for in the methods described here.

### 4.2.5 Other sources of error

Time-dependent phenomena not considered in our models may further bias age results. Constant production rate is an important assumption needed to simplify the nuclear build up process to apply a linear regression approach. In fact, the production rate is time dependent because the strength of Earth's magnetic field varies with time (Balco, 2017; Desilets et al., 2006; Dunai, 2001; Lifton et al., 2005; Stone, 2000). Constant inheritance is another key assumption. Sediments sampled from depth profiles are assumed to be well mixed at the time of deposition, such that the inherited concentration should be the same at every depth. This may not be true for sites with incremental deposition, and for sites where the depositional process or catchment-wide erosion rates vary with time.

### 5 Conclusions

In this paper, we introduce a combined least-squares inversion and Monte-Carlo approach to solve cosmogenic nuclide concentration depth profiles for surface exposure age and inheritance, considering erosion rate, erosion depth, muogenic production, and radioactive decay. Compared to existing models, our inversion approach offers a simple and direct way to estimate exposure ages, avoiding the non-linear effects of including the full muogenic production pathways where not warranted. In addition, this method allows propagation of all error sources using Monte-Carlo sampling to infer full probability distributions of age and inheritance.

Comparison of exposure ages estimated using our inverse models with the forward model of Hidy et al. (2010) confirms the robustness of our techniques, especially if using the eroded-thickness approach that includes muogenic production. The example of the Beida River T2 terraces shows that for sites with low inheritance, it is important include negative inheritance into the inversion process in order to fully characterize the distribution of the exposure age. Uncertainty analysis shows that the methods presented here yield suitable ages for surfaces with a total erosion thickness under two times attenuation length.



It is likely, however, that this level of erosion may not be well constrained and this ultimately sets a practical upper limit on the applicability of the CN technique for age-dating of depositional landforms.


## Appendix A

When r>0, the effective exposure age $T_e$ takes the following form,

$$T_{ei} = \left( \frac{1-e^{-\left(\frac{\rho r}{\Lambda_i}+\lambda\right)t}}{\frac{\rho r}{\Lambda_i}+\lambda} \right) = \frac{1-e^{-B_i t}}{B_i}, B_i = \frac{\rho r}{\Lambda_i} + \lambda, i = n, m1, m2. \quad (A1)$$

This suggests values for $T_e$ value are functions of both r and t for different production pathways. Between the two variables, r may be known, while t is unknown. Therefore, our aim is to rewrite A1 into an approximate form where t can be isolated. We first take a natural logarithm of the $T_{em}$ over $T_{en}$ ratio

$$ln\left(\frac{T_{em}}{T_{en}}\right) = ln\left(\frac{\frac{1-e^{-B_m t}}{B_m}}{\frac{1-e^{-B_n t}}{B_n}}\right) = ln\left(\frac{\frac{1-e^{-B_m t}}{B_m t}}{\frac{1-e^{-B_n t}}{B_n t}}\right) = ln\left(\frac{1-e^{-B_m t}}{B_m t}\right) - ln\left(\frac{1-e^{-B_n t}}{B_n t}\right) \quad (A2)$$

The expansion of a function, $f(x) = ln\left(\frac{1-e^{-x}}{x}\right)$, in A2 may be achieved by writing a Maclaurin series with the following form

$$f(x) = f(0) + f'(0)x + \frac{f''(0)}{2!}x^2 + \cdots \frac{f^{(k)}(0)}{k!}x^k + \cdots \quad (A3)$$

To write the expansion, we first rewrite f(x) as

$$f(x) = ln\left(\frac{1-e^{-x}}{x}\right) = ln\left(\frac{1-\left(1-x+\frac{x^2}{2!}-\frac{x^3}{3!}+\frac{x^4}{4!}-\cdots\right)}{x}\right) = ln\left(1-\frac{x}{2!}+\frac{x^2}{3!}-\frac{x^3}{4!}+\cdots\right) \quad (A4)$$

In this form, f(x) goes to zero when x goes to zero, therefore we have

$$f(0) = ln(1) = 0 \qquad (A5a)$$

$$f'(x) = \frac{-\frac{1}{2}+\frac{x}{3}-\frac{x^2}{8}+\cdots}{1-\frac{x}{2!}+\frac{x^2}{3!}-\frac{x^3}{4!}+\cdots} \text{ and } f'(0) = \frac{-\frac{1}{2}}{1} = -\frac{1}{2} \qquad (A5b)$$

$$f''(x)\left(1-\frac{x}{2!}+\frac{x^2}{3!}-\frac{x^3}{4!}+\cdots\right) + f'(x)\left(-\frac{1}{2}+\frac{x}{3}-\frac{x^2}{8}+\cdots\right) = \frac{1}{3}-\frac{x}{4}+\cdots \text{ and } f''(0) = \frac{1}{12} \quad (A5c)$$

We omit higher order derivatives from the series expansion.

$$\vdots$$

Bring A5 into A3, we have the expansion of f(x) as

$$f(x) = -\frac{x}{2} + \frac{x^2}{24} + O(x^3) \quad (A6)$$

Bringing A6 into A2, the natural logarithm of $T_{em}$ over $T_{en}$ ratio is



$$ln\left(\frac{T_{em}}{T_{en}}\right) = ln\left(\frac{\frac{1-e^{-B_mt}}{B_m}}{\frac{1-e^{-B_nt}}{B_n}}\right) = ln\left(\frac{1-e^{-B_mt}}{B_mt}\right) - ln\left(\frac{1-e^{-B_nt}}{B_nt}\right) = -\frac{B_mt}{2} + \frac{(B_mt)^2}{24} - \left(-\frac{B_nt}{2} + \frac{(B_nt)^2}{24}\right) + \left(O((B_mt)^3) - O((B_nt)^3)\right)$$

(A7)

The contribution of the third, $t^3$ term, is negligible and may be neglected. Making the substitution D = rt, the first-order terms in A7,

$$-\frac{B_mt}{2} + \frac{B_nt}{2} = -\frac{1}{2}\left(\frac{\rho D}{\Lambda_m} + \lambda t - \frac{\rho D}{\Lambda_n} - \lambda t\right) = -\frac{1}{2}\left(\frac{\rho D}{\Lambda_m} - \frac{\rho D}{\Lambda_n}\right).$$     (A8)

This result is independent of the exposure age, t.

With the same substitution, the second-order terms in A7,

$$\frac{(B_mt)^2}{24} - \frac{(B_nt)^2}{24} = \frac{1}{24}\left[\left(\frac{\rho D}{\Lambda_m} + \lambda t\right)^2 - \left(\frac{\rho D}{\Lambda_n} + \lambda t\right)^2\right] = \frac{1}{24}\left[\left(\frac{\rho D}{\Lambda_m}\right)^2 + \frac{2\rho D\lambda t}{\Lambda_m} + (\lambda t)^2 - \left(\frac{\rho D}{\Lambda_n}\right)^2 - \frac{2\rho D\lambda t}{\Lambda_n} - (\lambda t)^2\right] = \frac{1}{24}\left[\left(\frac{\rho D}{\Lambda_m}\right)^2 - \left(\frac{\rho D}{\Lambda_n}\right)^2\right] + \frac{1}{24}\left[\frac{2\rho D\lambda t}{\Lambda_m} - \frac{2\rho D\lambda t}{\Lambda_n}\right].$$     (A9)

In equation A9, the first term of the right-hand side is independent of age, t, while the second term is dependent of age. We therefore choose to omit the second term of equation A9 in order to develop an age-independent approximation. We find that this term may be omitted for two reasons. First, the absolute value of A8 is at least one order of magnitude larger than A9, therefore omitting one term from A9 will not lead to significant decrease of accuracy of the overall approximation. Second, for young surfaces, $\lambda t$ is sufficiently small that the second term of A9 is much smaller than the first term, which means omitting it will lead to even smaller decrease of accuracy.

Therefore, an approximate form of the eq. A7 that is independent of t is

$$ln\left(\frac{T_{em}}{T_{en}}\right) \approx -\frac{1}{2}\left(\frac{\rho D}{\Lambda_m} - \frac{\rho D}{\Lambda_n}\right) + \frac{1}{24}\left[\left(\frac{\rho D}{\Lambda_m}\right)^2 - \left(\frac{\rho D}{\Lambda_n}\right)^2\right]$$     (A10)

and the ratio between muon and nucleon effective age can be approximated as

$$\frac{T_{em}}{T_{en}} \approx e^{-\frac{1}{2}\left(\frac{\rho D}{\Lambda_m} - \frac{\rho D}{\Lambda_n}\right) + \frac{1}{24}\left[\left(\frac{\rho D}{\Lambda_m}\right)^2 - \left(\frac{\rho D}{\Lambda_n}\right)^2\right]}$$     (A11)

**Acknowledgments and data availability**

This work was supported by the US National Science Foundation [grant number EAR-1524734] to Michael Oskin, and through Cordell Durrell Geology Field Fund to Yiran Wang. We placed our code at GitHub (https://github.com/YiranWangYR/10BeLeastSquares).





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
