# Peer review of "Combined linear regression and Monte Carlo approach to modelling exposure age depth profiles"

_Geochronology, 2021_

## Referee Comment (RC2)

Review Summary:

In this paper, the authors present a novel approach to modelling depth profiles of 10Be using linear regression coupled with Monte Carlo sampling. This actually consists of two inversion approaches that consider mass-loss from the surface from different perspectives, erosion rate or net erosion, with the latter also including muon production with an approximation based on a Maclaurin series expansion of the classical depth profile equation. The authors test their model on datasets previously modelled by other published methods and demonstrate a reasonable replication of those results.

Generally, I think the linearization approach is very interesting, because it does simplify the age analysis from depth profiles into linear systems that are easier to manipulate. However, I do have some reservations about how the approximations might translate into biased probability density functions, and with some of the statements made regarding muon production. I also think more technical detail is needed on the model functionality and parameter specifics that include how sampling distributions are chosen. Additionally, there are several instances in the manuscript where statements are made that should be supported with citations. That said, the approach seems to have merit and I would be happy to recommend for publication provided the authors address the comments below.

Comments:

Line 19: I don't want to be a stickler on terminology, but I do recommend using TCN (terrestrial cosmogenic nuclide) vs. CN to separate this class of dating explicitly from extra-terrestrial applications.

Line 25-27: I completely agree with the difficulty. However, it should be mentioned that in some cases both can be ascertained. One would need the data resolution necessary to characterize the muon cross-over depth.

Line 37-38: It would seem this sentence should have a reference. To what linear inversion techniques are the authors referring?

Lines 40-42: v1.2 of the Hidy et al. 2010 calculator (released in 2012) is also Bayesian (see Mercader et al. 2012)

Lines 42-44: What is meant by stating that the available methods require prior knowledge of surface age and inheritance? Those are both free model parameters in the models I am familiar with. I think they may mean that some of those models require users to specify parameter boundaries—which should always be done arbitrarily large to avoid constraining the model. But that is not what is communicated here. What is communicated is that those models require some independent knowledge of those parameter values, which isn't the case.

Line 47: How is the minimum prior knowledge different between the linearization approach and others? Linear regression is functionally simpler, yes, but are the authors implying that their model has a reduction in degrees of freedom? What is the basis for this?

Line 65: Why use r vs. the more commonly used lowercase epsilon to represent erosion rate?

Lines 89-92: On one hand, yes muon production at the surface is small relative to spallogenic production, on the other hand it becomes increasingly important with depth. So, what does this mean for depth profiles where samples near the surface can't be obtained and muon production is far more important? This is a common issue, so should be addressed. Also, why ignore that 2%? Wouldn't it be a slightly better approximation to lump that 2% in with the nucleons and then treat it as simple exponential to linearize for the approximation?

Line 102: Agree. Also, there are lots of reference options that might be added here that support the benefit of constraining total eroded thickness.

Line 115: Should this reference actually be Braucher et al. (2009)? Also, this raises an interesting question…does the applied muon approximation approach offer at least the *possibility* of constraining a unique solution for age and erosion rate, or does that vanish with this approximation? This could be tested with a carefully composed pseudo-profile that characterizes the muon cross depth. I'd be more convinced of the acceptability of the approximation if the authors could show this. I'm still a bit concerned that there might be an issue here with deep profiles.

Line 127-130: What about uncertainty in density? I realize that this is basically an uncertainty in depth (assuming the authors are accounting for mass-depth), but it is unclear exactly how uncertainty in mass-depth is applied as it can include both a random (individual samples) and systematic component (effective depth shifting of all samples). Also, how does uncertainty in inheritance factor in at this stage?

Line 135: what corresponding probability density functions are used?

Line 137-139: how are the probability density functions calculated from the simulation results? Are the results weighted somehow, or is this a histogram vs. a pdf? It appears to be a histogram.

Table 3: In the Hidy et al. 2010 model of Lees Ferry, muons are not approximated with a two-term exponential (it uses a 5-term approximation like Schaller et al. 2002 and is internally optimized for the sample site and specific depth range). Also, the erosion rate range used was 0-0.4 cm/kyr. These differences should be noted.

Line 216: Where does the 0-0.32 cm/kyr erosion rate estimate come from?

Line 261-263: Not allowing negative inheritance actually changes the best-fit, or the peak in the distribution? I see how this would, and philosophically should, change the shape of the full distribution, but it shouldn't have an impact on the best fit—otherwise what makes it best? I guess it might because these are not probability density functions being generated, but histograms. So, doing this might actually be OK in the context of their modeling approach, but I'm hesitant to agree since I am unsure how all those allowable solutions with negative inheritance might introduce artefacts in other solution spaces.

Line 272-273: In the originally published Hidy et al (2010) Lees Ferry result, generous uncertainties in 10Be half-life (5%) and muon production (10%; probably still realistic considering Balco 2017) were

applied, so it would be useful to know what uncertainties were applied here for comparison. This could also explain some of the differences in results between those histograms and these.

Also, out of curiosity, I reran the original Lees Ferry dataset using the Bayesian version of the Hidy et al. (2010) model that generates actual probability density functions vs. the histograms of the original— basically by weighting all MC generated profiles (including solutions outside 95%) by the chi-squared likelihood function. Note that this is very different from what was presented in Hidy et al. (2010), but it is the version that has been adopted since 2012 so probably what should be used for a results comparison. With version 1.2, the results for age at 95% confidence are 76.6 – 96.1 ka (see figure below), with the probability weighting significantly tightening the distribution.

[Figure]

Lines 299-302: Yes, this can't be overstated! Also, there are numerous references out there that support the importance of soil processes to interpreting TCN profiles.

Lines 310-314: This is an interesting exercise, but are there approaches that ignore radioactive decay? Strange if there is.

Lines 330-332: Generally, I agree with this, but there are instances where dating highly eroded surfaces are useful when one is more interested in soil age vs. deposition age.

Lines 368-370: True, this isn't really a revelation though and is why many depth profiles end up reported with zero-erosion rate minimum ages when constraints on surface erosion can't be justified.

Lines 399-401: I disagree with this statement. While it may be true for this modeling approach, it is incorrect to infer for all inversion models that may apply different statistical methods for reporting solutions.

---

## Editor Comment (EC1)

**Review of "Combined linear regression and Monte Carlo approach to modelling exposure age depth profiles" by Yiran Wang and Michael E. Oskin**

Pieter Vermeesch
University College London
p.vermeesch@ucl.ac.uk

January 26, 2022

First of all, I would like to apologise for the lengthy review process. Your paper is quite technical and it was not so easy to find qualified reviewers for it. Unfortunately, most of those qualified reviewers are very busy. I am still awaiting one more review, but in anticipation of this, I will already share my own thoughts on your manuscript.

In your paper, you present an improved method to fit cosmogenic nuclide depth profiles. As explained in the introduction, there currently are two ways to do so:

1. Linear regression, ignoring muons.

2. Iterative forward-inverse models, including muons.

The manuscript under consideration presents a hybrid approach, in which the linear regression approach is modified to account for muons, and Monte Carlo simulations are used to estimate uncertainties.

**1 Linear fitting with muons**

The manuscript shows that, if the thickness of the eroded layer ($D$) is known, it is possible to compute a depth-dependent 'effective production rate' that accounts for both the neutron and the (slow+fast) muons. This is achieved using a second order Taylor series approximation. That clever trick is the main novelty in this paper.

One problem with the new method is that it requires prior knowledge of the eroded thickness, which violates one of the principles set out in the introductory sections of the paper. According to lines 35–36 of the manuscript, the main advantage of the linear regression approach to depth-profile modelling is that it can "determine an exposure age without any prior knowledge". The second order Taylor approximation trick has removed this advantage. Then why not switch to iterative methods entirely?

In other words, I must agree with the first reviewer that it is not clear what value the new method adds to the cosmogenic nuclide toolbox. Perhaps your method does have advantages, but these are not clear to me at the moment. Probably the best way to demonstrate the alleged superiority of the proposed method is with synthetic examples. Can you come up with a realistic scenario in which the modified linear regression will provide better (i.e. more accurate and/or precise) results than the iterative method?

**2 Monte Carlo error estimation**

The paper presents an ad-hoc algorithm to estimate the analytical uncertainties of their exposure age estimates by 'Monte Carlo simulation'. My understanding is that this algorithm works as follows:

1. Given a number of measurements and parameters, fit the data by least squares regression of Equation 10. Then calculate the exposure age from that fit using Equation 11.

2. Create new parameter values by drawing random numbers from user-specified prior distributions. Similarly, create new data by drawing random numbers from normal distributions with means corresponding to the measurements, and standard deviations given by the analytical uncertainties.

3. Refit the data.

4. Repeat steps 2 and 3 a few thousand times and investigate the distribution of fit parameters.

This method seems problematic to me because it mixes two kinds of uncertainties:

1. Parameter uncertainties such as the thickness of the eroded layer.

2. Analytical uncertainty of the $^{10}$Be measurements.

I'm not sure if you can treat these two sources of uncertainty equally. Parameter uncertainties represent some kind of prior belief about the plausible true values. In contrast, analytical uncertainties represent the standard error of the mean given a finite number of measurements, where the mean is offset from the true value by an unknown amount.

Instead of splitting the calculation into two steps (regression & estimation), parameter estimation could also be done in one step by replacing Equation 11 with the following log-likelihood function:

$$LL(t, C_{inh}, D | C(z), s[C(z)]) = \text{Constant} - \frac{1}{2} \sum_{j=1}^{n} \left( \frac{C(z_j) - P_e(z_j) T_{en}(t, D) - C_{inh}}{s[C(z_j)]} \right)^2$$

where $z = \{z_1, z_2, ..., z_n\}$, $P_e(z_j)$ stands for $P_{ze}$ in the manuscript (at depth $z_j$), and $T_{en}(t, D)$ is shorthand for Equation 8 (with $i = n$). The 'best' values for $t$, $C_{inh}$ and $D$ are the ones that maximise $LL$. The (co)variance (matrix) of $t$ and $C_{inh}$ can then be estimated by inverting the matrix of second derivatives of $-LL$ with respect to $t$, $C_{inh}$ and $D$ at the maximum likelihood values.

An alternative approach is to use the log-likelihood function in a Bayesian framework, e.g. using the Metropolis-Hastings algorithm. You can then assign your preferred probability distribution for the parameters as 'prior information'. MCMC modelling will yield posterior distributions for $t$, $C_{inh}$ and $D$ that will look somewhat similar to (but not exactly the same as) Figures 2–5 of the manuscript.

Of course, both of these approaches effectively remove the key characteristic of your linear regression. So, in effect, it turns your hybrid approach to depth profile modelling into a standard forward-inverse modelling approach.

**3 Negative inheritance**

Lines 177–178 of the manuscript state that "some inversion results yield non-physical predictions with negative inheritance. These negative inheritance predictions are necessary to estimate the full distribution of the exposure age, but we exclude these from the final inheritance results."

This does not make sense. An inversion result that yields non-physical predictions is wrong. A pragmatic way to fix this issue is to parameterise your problem in terms of the logarithm of inheritance, rather than inheritance itself. After the optimisation, you can then exponentiate the maximum likelihood value to ensure a strictly positive result.

**4    Scaling models**

Your method is based on an exponential approximation for the depth-dependence of muon production, but as far as I can tell, the parameters of the exponential approximation are not varied with elevation or location. This is incorrect. At shallow depths the absolute magnitude of the muon production rate is quite variable with elevation, so even if you use an exponential approximation the value for the surface production rate for each reaction would have to be different at each site. This could be easily fixed by using a more complicated muon production model at the beginning to compute site-specific exponentials for each site. But without fixing this, the age estimates are expected to be fairly inaccurate in lots of cases.

---

## Author Comment (AC2)

Reply to the 1st reviewer:

*Despite being in a field I really appreciate, I have some difficulties to judge what is the value added by this paper. This is probably by the fact that too many assumptions or to be more precise too many shortcuts are used to simplify the main equation governing the cosmogenic production equation as a function of denudation rate and time. Some of these shortcuts are dangerous and some other can be avoided with the used of numerical calculations. I will thus ask for a revision of this paper*

*In the entire paper I suggest changing erosion by denudation that is more appropriate for cosmo.*

Thank you for the suggestion, we will change that in our revision.

*At the end of abstract you mention "compared to the error from omitting muogenic production…" I totally agree so, why do you present a linearization that ignores muons contributions?*

The formative paper using linear regression to interpret depth profiles omits muogenic production (Anderson et al., 1996) so it is important to at least comment on the error that arises from their omission.

*Line 35-40: despite muons contributions are small at surface compared to the neutron one, ignoring their contributions and considering only neutrons will yield to multiple time/denudation pairs that can model a depth distribution.*

We agree with the reviewer.

*Line 44: If you want to be totally objective you should live all parameters free and in a second step consider the solutions that can match the field observations. If you constrain at the first step your unknowns, time or denudation you may miss the real solutions.*

We realise the claim of not requiring any prior knowledge in the introduction is inappropriate. What we meant is linear regression method doesn't require prior knowledge of the exposure age and inheritance, while the erosion rate (or eroded thickness) is a required prior knowledge for a single isotope depth profile method. We will revise the introduction and focus the motivation of our study to expanding the use of the linear inversion method in exposure-age dating.

*Line 55 Legend of Figure 1: you should update the muon contributions; since Braucher 2003, these contribution have been updated (Braucher 2011,2013, Balco 2008, 2017 ). More it has been also shown that Heisinger muons contributions were too high. You should correct them in your matlab code and in the Hidy one.*

Thank you for the suggestion. We will correct this in our revision.

*Line 90-91: again do not omit muon contributions! In a high denudation environment, their contributions are far from being negligible.*

We agree. We include this to make comparisons later in the discussion,

*Line 67: I think Nishiizumi, 2007 is not appropriate as in this paper he proposed a half-life of 1.36±0.07 Ma.*

Yes, we will take this reference off from our revision.

*Line 100 and following paragraph: I think this is not the right approach. First I will have a look to the distribution as a function of depth (in g/cm2) to see within the first two meters what is the value of the "slope" of the exponential decrease. Lower than 250 g/cm2 will traduce a contribution mainly due to neutrons with moderate denudation rate. If higher muon contributions are more important due to higher denudation rate or can be due to a recent rejuvenation of the profile making deep samples to be now closer to surface. In this latter case, running an inversion model with density as free parameter will probably propose high values for density >3 g/cm3 making clear that the profile has been perturbed. This can be the case when loess covers are rapidly eroded by wind deflation, so fast that the cosmo production cannot be at equilibrium.*

*Therefore I will let run the model with totally free parameters and then cut the Time/ denudation space by probable eroded thickness to reduce this space. By imposing since the beginning of the modelling a constrain as important as the eroded thickness may be dangerous to my point of view.*

Thank you for the suggestion. In this paper, we are attempting to provide an approach for surfaces under constant erosion, therefore we exclude any abrupt change of the deposition/erosion environment from the model.

*Line 108: which muon contribution do you used as Tem? Fast or slow? Is this choice important?*

We use both, as shown in eq. 9. We will revise this sentence to make it clear

*Please change the * by × in the tables. Please use uniform values for concentraions ( at/g or 105at/g )*

We will correct this in our revision

*Line 174: why this denudation rate of 0.3±0.05 cm/kyr ?*

This rate is calculated based on the 40±10 cm denudation and the surface age estimated using the eroded thickness approach. We will clarify this in our revision

*With this loess covered surface, probably the use of two nuclides will be better than one.*

Thank you for bringing this important point up. The loess deposit is well-dated, young, and quite continuous. Therefore, the concentration before loess deposition can be easily modelled. Using two nuclides would offer an additional constraint, but this is beyond the scope of this manuscript.

*Line 177: I am not convinced by the fact that you authorize inheritance to be negative. This is as you mentioned "non-physical". Therefore what will happen if you restrict the modelling to inheritance ≥ to zero? Is the overall space of solution affected?*

Yes, the overall space of solution will be affected if negative inheritance estimations are omitted. For example, suppose we have a sample group with a true age of 100 kyr and a true inheritance of 0 atoms/g. Because of the uncertainties within the sample measurements, ideally, we want to have the estimated distribution of the exposure age to be centered around 100 kyr, with approximately half of the estimation older than 100 kyr, the other half younger than 100 kyr. The estimations younger than 100 kyr correlate to some positive inheritance, while those older than 100 kyr correlate to negative inheritance. The overall distributions of the inheritance should be centered around 0, meaning that approximately half of the estimated inheritance should be negative. If we require non-negative inheritance during the inversion, it will end up with deviated estimation results: the inheritance would center around a positive value, while the central age would be younger than 100 kyr.

We tested the two methods (least square linear inversion described in our manuscript and a Metropolis-Hastings Monte Carlo Bayesian inversion) using pseudo depth profiles with zero inheritance. We find that requiring non-negative inheritance during inversion leads to underestimation of the exposure age for both approaches. We are considering including this into the revision.

*Paragraph 4.2.5 : I agree but using variable production rates implies adding more uncertainties and this is not the fact in the actual calculators !*

We agree with the reviewer. We will clarify this in the revision

*If you think to revise this contribution you should try to add a second nuclides (26Al for example) and try to remodel the depth profile with two nuclides. Inheritance can thus be variable and this can probably be a great value to the modelling of depth profile.*

Thank you for the suggestion. We take our attempt with the 10Be profile as a first step and would like to perfect it before we venture into the terrane of multiple nuclides. We will definitely consider bring in a second nuclide in the future.

---

## Author Comment (AC3)

*Comments: Line 19: I don't want to be a stickler on terminology, but I do recommend using TCN (terrestrial cosmogenic nuclide) vs. CN to separate this class of dating explicitly from extra-terrestrial applications.*

Thank you for the suggestion. We will change the terminology in our revision

*Line 25-27: I completely agree with the difficulty. However, it should be mentioned that in some cases both can be ascertained. One would need the data resolution necessary to characterize the muon crossover depth.*

Thank you for the suggestion, we will add this to the revision.

*Line 37-38: It would seem this sentence should have a reference. To what linear inversion techniques are the authors referring?*

The linear inversion technique was introduced by Anderson et al., 1996. We will add the reference to the sentence.

*Lines 40-42: v1.2 of the Hidy et al. 2010 calculator (released in 2012) is also Bayesian (see Mercader et al. 2012)*

We apologize for this mistake; we will correct it in our revision.

*Lines 42-44: What is meant by stating that the available methods require prior knowledge of surface age and inheritance? Those are both free model parameters in the models I am familiar with. I think they may mean that some of those models require users to specify parameter boundaries—which should always be done arbitrarily large to avoid constraining the model. But that is not what is communicated here. What is communicated is that those models require some independent knowledge of those parameter values, which isn't the case.*

We realise the claim of not requiring any prior knowledge in the introduction is inappropriate. What we meant is linear regression method inverse for exposure age and inheritance directly, without any pre-set boundaries. We agree that theoretically the boundaries can be set arbitrarily large to avoid constraining the model, while in practice, we find a certain degree of prior knowledge is useful to save model running time, or to avoid parameters been trapped in unrealistic solution space.

As an inverse approach, the least squares linear regression approach directly solves for age and inheritance, while treating the erosion rate/eroded thickness as an input instead of an output of the model. This characteristic makes it a convenient tool for exposure age estimation. It can be used with Monte Carlo sampling to explore the full distribution of possible ages and inheritance from the variation of input parameters (including erosion). Linear regression is also useful as a starting point for forward (e.g. Bayesian) models. Inverse-modelled age and inheritance may thus help researchers to tune the boundary values for the forward models to get better simulation results. Therefore, instead of replacing forward methods, we will argue that our approach complements forward methods.

*Line 47: How is the minimum prior knowledge different between the linearization approach and others? Linear regression is functionally simpler, yes, but are the authors implying that their model has a reduction in degrees of freedom? What is the basis for this?*

The degrees of freedom are the same for both methods.

*Line 65: Why use r vs. the more commonly used lowercase epsilon to represent erosion rate?*

We will change it to ε instead.

*Lines 89-92: On one hand, yes muon production at the surface is small relative to spallogenic production, on the other hand it becomes increasingly important with depth. So, what does this mean for depth profiles where samples near the surface can't be obtained and muon production is far more important? This is a common issue, so should be addressed. Also, why ignore that 2%? Wouldn't it be a slightly better approximation to lump that 2% in with the nucleons and then treat it as simple exponential to linearize for the approximation?*

We agree. We address the effect of muons in our discussion section. The erosion rate approach, excluding muons, is useful for exploring some of the trade-offs between erosion rate and age (using Te, eq. 7). We will rearrange our manuscript to make this point clear.

*Line 102: Agree. Also, there are lots of reference options that might be added here that support the benefit of constraining total eroded thickness.*

We will include references here in the revision.

*Line 115: Should this reference actually be Braucher et al. (2009)? Also, this raises an interesting question…does the applied muon approximation approach offer at least the possibility of constraining a unique solution for age and erosion rate, or does that vanish with this approximation? This could be tested with a carefully composed pseudo-profile that characterizes the muon cross depth. I'd be more convinced of the acceptability of the approximation if the authors could show this. I'm still a bit concerned that there might be an issue here with deep profiles.*

We apologize for the mistake; we will correct it in our revision.

Whether calculated using erosion rate or eroded thickness, our least-squares approach requires external information (or assumption) for the erosion rate/thickness, therefore cannot be used to calculate a unique solution for age and erosion rate.

We will include an example in the discussion to show how our approach behaves with deep profiles.

*Line 127-130: What about uncertainty in density? I realize that this is basically an uncertainty in depth (assuming the authors are accounting for mass-depth), but it is unclear exactly how uncertainty in mass depth is applied as it can include both a random (individual samples) and*

*systematic component (effective depth shifting of all samples). Also, how does uncertainty in inheritance factor in at this stage?*

We didn't mention uncertainty in density, but we have considered it in our approach. We will add this to the revision. The uncertainty in inheritance is the outcome of the simulation.

*Line 135: what corresponding probability density functions are used?*

Either uniform or normal distributions. We will clarify this in our revision.

*Line 137-139: how are the probability density functions calculated from the simulation results? Are the results weighted somehow, or is this a histogram vs. a pdf? It appears to be a histogram.*

We currently only present the results in histograms. We will clarify that in our revision. We are also considering adding estimation of the pdf into our codes.

*Table 3: In the Hidy et al. 2010 model of Lees Ferry, muons are not approximated with a two-term exponential (it uses a 5-term approximation like Schaller et al. 2002 and is internally optimized for the sample site and specific depth range). Also, the erosion rate range used was 0-0.4 cm/kyr. These differences should be noted.*

Thank you for pointing this out, we will clarify that in our revision.

*Line 216: Where does the 0-0.32 cm/kyr erosion rate estimate come from?*

We find that using the full range of erosion rate (0 to 0.4 cm/kyr) results in an age range that is too old and inconsistent with the total erosion of 0 to 30 cm. This is because the upper bound on total erosion limits the range of acceptable results in the Hidy et al. (2010) model. Because our least-squares approach uses either erosion rate or total eroded thickness, but not both, we chose to set a narrower limit on the erosion rate, from 0-0.32 cm/kyr, so that the ages and total eroded thickness were consistent with that cited in the original paper. Using our eroded thickness fitting approach also avoids this problem. We will clarify this point in the revised paper.

*Line 261-263: Not allowing negative inheritance actually changes the best-fit, or the peak in the distribution? I see how this would, and philosophically should, change the shape of the full distribution, but it shouldn't have an impact on the best fit—otherwise what makes it best? I guess it might because these are not probability density functions being generated, but histograms. So, doing this might actually be OK in the context of their modeling approach, but I'm hesitant to agree since I am unsure how all those allowable solutions with negative inheritance might introduce artefacts in other solution spaces.*

We will be specific that the negative inheritance effect is important for our approach. We will also include a comparison of inversion results using pseudo profiles with and without negative inheritance in our revision.

*Line 272-273: In the originally published Hidy et al (2010) Lees Ferry result, generous uncertainties in 10Be half-life (5%) and muon production (10%; probably still realistic considering Balco 2017) were applied, so it would be useful to know what uncertainties were applied here for comparison. This could also explain some of the differences in results between those histograms and these. Also, out of curiosity, I reran the original Lees Ferry dataset using the Bayesian version of the Hidy et al. (2010) model that generates actual probability density functions vs. the histograms of the original— basically by weighting all MC generated profiles (including solutions outside 95%) by the chi-squared likelihood function. Note that this is very different from what was presented in Hidy et al. (2010), but it is the version that has been adopted since 2012 so probably what should be used for a results comparison. With version 1.2, the results for age at 95% confidence are 76.6 – 96.1 ka (see figure below), with the probability weighting significantly tightening the distribution.*

The version 1.2 we had gave us a resulting age range of 74.5 – 98.7 ka (we used the predefined Lees Ferry settings that came along with the program). We are not sure the source of this discrepancy. We are happy to use the new age range you present here, and we will consider the differences in parameter value and parameter uncertainties in our revision.

*Lines 299-302: Yes, this can't be overstated! Also, there are numerous references out there that support the importance of soil processes to interpreting TCN profiles.*

We will include references in our revision.

*Lines 310-314: This is an interesting exercise, but are there approaches that ignore radioactive decay? Strange if there is.*

We will remove this section from the revision.

*Lines 330-332: Generally, I agree with this, but there are instances where dating highly eroded surfaces are useful when one is more interested in soil age vs. deposition age.*

We agree. We should be specific that we do not recommend our approach to date surfaces with large eroded thicknesses.

*Lines 368-370: True, this isn't really a revelation though and is why many depth profiles end up reported with zero-erosion rate minimum ages when constraints on surface erosion can't be justified.*

We agree. However, we feel this is important to state and will keep this sentence as written.

*Lines 399-401: I disagree with this statement. While it may be true for this modeling approach, it is incorrect to infer for all inversion models that may apply different statistical methods for reporting solutions.*

We will be specific that the negative inheritance effect is important for our approach.

---

## Author Response (AR1)

**Reply to comments**

Dear editor and reviewers,

I have the pleasure of sending you the revised manuscript entitled "Combined linear regression and Monte Carlo approach to modelling exposure age depth profiles" authored by Yiran Wang and Michael E. Oskin to be considered for publication as a research article in Geochronology.

Fully considering the concerns of the editor and the reviewers, we have made some major revision of this manuscript.

- We have rewritten the introduction, focusing our motivation on expanding the application of linear regression approach, and on its advantages: easy to invert, and provides a direct way to explore the trade-offs between erosion and age.

- We have included a new section, "Simulated depth profiles" (section 3.1), which demonstrates the performance of our approach under various circumstances. We included scenarios such as low inheritance, large denudation, and deep profiles, to address the concerns raised by the editor and reviewers.

- We have deleted demonstration of the erosion/denudation-rate approach for the case examples to shorten the manuscript, while kept demonstration of the more accurate denudation-length approach (section 3.2). We have also added a demonstration of using the denudation-rate approach to explore the denudation-rate vs exposure age relationship for the Beida River site (L263-268).

- We have made other minor revisions, including using site specific muon production rates for our models; details are in our point-by-point replies.

We appreciate your consideration of our manuscript, and we look forward to receiving further comments.

Yiran Wang

Reply to the editor, Dr. Pieter Vermeesch:

1. *Linear fitting with muons.*

We agree with the reviewers that it was inappropriate to claim that our method does not require any prior knowledge: erosion depth or rate must be estimated. What we meant to convey is that linear regression inverts for exposure age and inheritance directly, without needing to define a pre-set boundary or initial prior distribution for these. The erosion rate (or eroded thickness) is a required input for our inverse modelling approach.

Our motivation for introducing this approach is, first, to expand the application of linear regression to model exposure-age depth profiles. Prior published linear regression methods ignore muons and do not adequately address erosion. Our approach incorporates muogenic production, making it possible to estimate ages for sites with steady rates or a defined amount of erosion. As demonstrated in the discussion section, this method can provide exposure age estimations with high precision for profiles with less than 2 times attenuation length erosion, suggesting it may be applied for most surface exposure dating scenarios. Our second motivation is that as an inverse approach, the least squares linear regression approach directly solves for age and inheritance, while treating the erosion rate/eroded thickness as an input instead of an output of the model. This characteristic makes it a convenient tool for exposure age estimation. It can be used with Monte Carlo sampling to explore the full distribution of possible ages and inheritance from the variation of input parameters (including erosion). Linear regression is also useful as a starting point for forward (e.g. Bayesian) models. Inverse-modelled age and inheritance may thus help researchers to tune the boundary values for the forward models to get better simulation results. Therefore, instead of replacing forward methods, we will argue that our approach complements forward methods. In addition, as an added benefit of our inverse model approach, the effective age Te provides a straightforward way to explore the trade-off between erosion rate and exposure age.

We have incorporated these ideas into our revised manuscript, and completely rewrote the introduction.

By comparing results from our least squares linear regression with a Bayesian approach using Metropolis-Hastings sampling (Section 3.1 of the revised manuscript), we find the age estimations from linear regression have precision and accuracy comparable to results from Bayesian approach, under various circumstances. These examples support our claim that the linear regression approach is robust.

2. *Monte Carlo error estimation*

The editor's description of the procedure of our inversion and Monte Carlo simulation approach is correct. We would like to modify the procedure as follows:

1. Generate all the input parameters (10Be concentration, sample depth, eroded thickness, production rate, attenuation length, etc.) from pre-defined distributions.
2. Fit the data with eq. 10 to get Ten and inheritance.
3. Calculate the exposure age use Ten from step 2 and parameters generated from step 1 using eq. 11.
4. Repeat step 1-3 for many times, sampling the underlying distributions of each parameter, to produce a distribution of results.

In our inversion, we treat the erosion as an input of the model, therefore the uncertainty related to the erosion can be treated the same way as other parameters. We treat the erosion as a known value instead of an inversion output for several reasons. First, it simplifies the inversion process, so that one can inverse linearly in 2D space, and the Te value resulted from the first step could be used to explore the trade-off between erosion rate and exposure age. Second, though theoretically it is possible to find a unique set of solutions for erosion rate and exposure age for a certain depth profile, in practice, the sample uncertainties always exceed the resolution that required to define that unique solution. This means including erosion as an unknown is unlikely to increase the utility of the estimation results. As mentioned earlier, we are considering including comparisons of our least squares linear inversion and Bayesian inversion with pseudo depth profiles to demonstrate that a simple Monte Carlo sampling with two-step estimation can provide sufficiently accurate results.

In the revision, we modified our sampling steps (L140-145) to make it clearer. We also emphasised in the introduction (L33-34) about treating erosion as a model input.

**3. Negative Inheritance**

Negative inheritance estimations can be prevented from inversion process, and we have incorporated that into our earlier published work (e.g. Wang et al., 2020). However, we realized in the writing of this manuscript that those negative results, though physically impossible, are necessary for mathematical reasons.

For example, suppose we have a sample group with a true age of 100 kyr and a true inheritance of 0 atoms/g. Because of the uncertainties within the sample measurements, ideally, we want to have the estimated distribution of the exposure age to be centered around 100 kyr, with approximately half of the estimation older than 100 kyr, the other half younger than 100 kyr. The estimations younger than 100 kyr correlate to some positive inheritance, while those older than 100 kyr correlate to negative inheritance. The overall distributions of the inheritance should be centered around 0, meaning that approximately half of the estimated inheritance should be negative. If we require non-negative inheritance during the inversion, it will end up with deviated estimation results: the inheritance would center around a positive value, while the central age would be younger than 100 kyr.

To demonstrate this, we tested the two methods (least squares linear inversion described in our manuscript and a Metropolis-Hastings Monte Carlo Bayesian inversion) using pseudo depth profiles with true inheritance equals to zero and 5000 atoms/g. We find that not permitting negative inheritance during inversion leads to significant underestimation of the exposure age for both approaches. For cases with low but not zero (5000 atoms/g), we find the underestimation still occurs, but not as significant as the zero-inheritance case. (Section 3.1.2).

**4. Scaling methods**

Thank you for the suggestion. We updated muon production rate used in the pseudo profiles and in the Beida River case. We also deleted the sentences that indicate fixed muon production rate in our revised manuscript.

Reply to the 1st reviewer:

*Despite being in a field I really appreciate, I have some difficulties to judge what is the value added by this paper. This is probably by the fact that too many assumptions or to be more precise too many shortcuts are used to simplify the main equation governing the cosmogenic production equation as a function of denudation rate and time. Some of these shortcuts are dangerous and some other can be avoided with the used of numerical calculations. I will thus ask for a revision of this paper*

*In the entire paper I suggest changing erosion by denudation that is more appropriate for cosmo.*

Thank you for the suggestion, we have changed erosion by denudation in our revision.

*At the end of abstract you mention "compared to the error from omitting muogenic production…" I totally agree so, why do you present a linearization that ignores muons contributions?*

The formative paper using linear regression to interpret depth profiles omits muogenic production (Anderson et al., 1996) so it is important to at least comment on the error that arises from their omission. We also find the erosion rate approach (omitting muons) is helpful in demonstrating the trade-offs between denudation and exposure age. We clarified these points in the abstract (L14-15) and the introduction (L41-44) in our revision.

*Line 35-40: despite muons contributions are small at surface compared to the neutron one, ignoring their contributions and considering only neutrons will yield to multiple time/denudation pairs that can model a depth distribution.*

We agree with the reviewer. We have completely rewritten the introduction to address the concerns of the editor and reviewers.

*Line 44: If you want to be totally objective you should live all parameters free and in a second step consider the solutions that can match the field observations. If you constrain at the first step your unknowns, time or denudation you may miss the real solutions.*

We realise the claim of not requiring any prior knowledge in the introduction is inappropriate. What we meant is linear regression method doesn't require prior knowledge of the exposure age and inheritance, while the erosion rate (or eroded thickness) is a required prior knowledge for a single isotope depth profile method. We have completely rewritten the introduction and focused the motivation of our study to expanding the use of the linear inversion method in exposure-age dating.

*Line 55 Legend of Figure 1: you should update the muon contributions; since Braucher 2003, these contribution have been updated (Braucher 2011,2013, Balco 2008, 2017 ). More it has been also shown that Heisinger muons contributions were too high. You should correct them in your matlab code and in the Hidy one.*

Thank you for the suggestion. In the revision, we use the SLHL production rate based on Martin et al., 2017 and Balco 2017.

*Line 90-91: again do not omit muon contributions! In a high denudation environment, their contributions are far from being negligible.*

We agree. We find the erosion rate approach (omitting muons) is helpful in demonstrating the trade-offs between denudation and exposure age. We clarified these points in the introduction (L41-44) in our revision and demonstrate the application in the case examples (section 3.2, L263-269). In the discussion, we also explored the error related to omit muon (section 4.2.2)

*Line 67: I think Nishiizumi, 2007 is not appropriate as in this paper he proposed a half-life of 1.36±0.07 Ma.*

Yes, we took this reference off from our revision.

*Line 100 and following paragraph: I think this is not the right approach. First I will have a look to the distribution as a function of depth (in g/cm2) to see within the first two meters what is the value of the "slope" of the exponential decrease. Lower than 250 g/cm2 will traduce a contribution mainly due to neutrons with moderate denudation rate. If higher muon contributions are more important due to higher denudation rate or can be due to a recent rejuvenation of the profile making deep samples to be now closer to surface. In this latter case, running an inversion model with density as free parameter will probably propose high values for density >3 g/cm3 making clear that the profile has been perturbed. This can be the case when loess covers are rapidly eroded by wind deflation, so fast that the cosmo production cannot be at equilibrium.*

*Therefore I will let run the model with totally free parameters and then cut the Time/ denudation space by probable eroded thickness to reduce this space. By imposing since the beginning of the modelling a constrain as important as the eroded thickness may be dangerous to my point of view.*

Thank you for the suggestion. In this paper, we are attempting to provide an approach for surfaces under constant erosion, therefore we exclude any abrupt change of the deposition/erosion environment from the model, except where such a change may be independently modelled and removed, as is the case for the loess cover at the Beida River site (L258).

*Line 108: which muon contribution do you used as Tem? Fast or slow? Is this choice important?*

We use both, as shown in eq. 9. We revised this sentence as "Using a series expansion, we rewrite the effective age related to muons (negative and fast), $T_{em}$" to make it clear (L104).

*Please change the * by × in the tables. Please use uniform values for concentraions ( at/g or 105at/g )*

We corrected this in our revision.

*Line 174: why this denudation rate of 0.3±0.05 cm/kyr ?*

This rate is calculated based on the 40±10 cm denudation and the surface age estimated using the eroded thickness approach. In the revision, we have removed this part from the manuscript and only demonstrated the most accurate denudation-depth approach.

*With this loess covered surface, probably the use of two nuclides will be better than one.*

Thank you for bringing this important point up. The loess deposit is well-dated, young, and quite continuous. Therefore, the concentration before loess deposition can be easily modelled. Using two nuclides would offer an additional constraint, but this is beyond the scope of this manuscript.

*Line 177: I am not convinced by the fact that you authorize inheritance to be negative. This is as you mentioned "non-physical". Therefore what will happen if you restrict the modelling to inheritance ⩾ to zero? Is the overall space of solution affected?*

Yes, the overall space of solution will be affected if negative inheritance estimations are omitted. See reply to editor comment, above. We have also included a simulation in section 3.1.2 to show this effect, and included a more detailed discussion in L386-397.

*Paragraph 4.2.5 : I agree but using variable production rates implies adding more uncertainties and this is not the fact in the actual calculators !*

We agree with the reviewer. We change the sentence as "In fact, the production rate is time dependent because the strength of Earth's magnetic field varies with time (Balco, 2017; Desilets et al., 2006; Dunai, 2001; Lifton et al., 2005; Stone, 2000). Extending our model to account for temporally variable production rate is beyond the scope of present study." (L400-402)

*If you think to revise this contribution you should try to add a second nuclides (26Al for example) and try to remodel the depth profile with two nuclides. Inheritance can thus be variable and this can probably be a great value to the modelling of depth profile.*

Though a second nuclide would certainly add to the available constraints, this is beyond the scope of our manuscript, which is focussed on presenting a least-squares solution to single-nuclide measurements.

Reply to Dr. Alan Hidy

*Comments: Line 19: I don't want to be a stickler on terminology, but I do recommend using TCN (terrestrial cosmogenic nuclide) vs. CN to separate this class of dating explicitly from extra-terrestrial applications.*

Thank you for the suggestion. We have changed the terminology in our revision.

*Line 25-27: I completely agree with the difficulty. However, it should be mentioned that in some cases both can be ascertained. One would need the data resolution necessary to characterize the muon crossover depth.*

We agree. We have completely rewritten our introduction to focus on expanding the linear regression approach to include erosion and muons.

*Line 37-38: It would seem this sentence should have a reference. To what linear inversion techniques are the authors referring?*

The linear inversion technique was introduced by Anderson et al., 1996. We have added the reference to the sentence.

*Lines 40-42: v1.2 of the Hidy et al. 2010 calculator (released in 2012) is also Bayesian (see Mercader et al. 2012)*

We apologize for this mistake; we have generalized the referenced approached as forward-modeling approaches instead in our revision (L37).

*Lines 42-44: What is meant by stating that the available methods require prior knowledge of surface age and inheritance? Those are both free model parameters in the models I am familiar with. I think they may mean that some of those models require users to specify parameter boundaries—which should always be done arbitrarily large to avoid constraining the model. But that is not what is communicated here. What is communicated is that those models require some independent knowledge of those parameter values, which isn't the case.*

We realise the claim of not requiring any prior knowledge in the introduction is inappropriate. What we meant is linear regression method inverse for exposure age and inheritance directly, without any pre-set boundaries. We agree that theoretically the boundaries can be set arbitrarily large to avoid constraining the model, while in practice, we find a certain degree of prior knowledge is useful to save model running time, or to avoid parameters been trapped in unrealistic solution space.

As an inverse approach, the least squares linear regression approach directly solves for age and inheritance, while treating the erosion rate/eroded thickness as an input instead of an output of the model. This characteristic makes it a convenient tool for exposure age estimation. It can be used with Monte Carlo sampling to explore the full distribution of possible ages and inheritance from the variation of input parameters (including erosion). Linear regression is also useful as a starting point

for forward (e.g. Bayesian) models. Inverse-modelled age and inheritance may thus help researchers to tune the boundary values for the forward models to get better simulation results. Therefore, instead of replacing forward methods, we argue that our approach complements forward methods.

We have incorporated these ideas into our revised manuscript, and completely rewrote the introduction.

*Line 47: How is the minimum prior knowledge different between the linearization approach and others? Linear regression is functionally simpler, yes, but are the authors implying that their model has a reduction in degrees of freedom? What is the basis for this?*

The degrees of freedom are the same for both methods.

*Line 65: Why use r vs. the more commonly used lowercase epsilon to represent erosion rate?*

We have changed it to ε instead.

*Lines 89-92: On one hand, yes muon production at the surface is small relative to spallogenic production, on the other hand it becomes increasingly important with depth. So, what does this mean for depth profiles where samples near the surface can't be obtained and muon production is far more important? This is a common issue, so should be addressed. Also, why ignore that 2%? Wouldn't it be a slightly better approximation to lump that 2% in with the nucleons and then treat it as simple exponential to linearize for the approximation?*

We agree. We address the effect of muons in our discussion section. The erosion rate approach, excluding muons, is useful for exploring some of the trade-offs between erosion rate and age (using Te, eq. 7). We have added this clarification in the revised introduction (L41-44).

*Line 102: Agree. Also, there are lots of reference options that might be added here that support the benefit of constraining total eroded thickness.*

We have included references here in the revision (L99).

*Line 115: Should this reference actually be Braucher et al. (2009)? Also, this raises an interesting question…does the applied muon approximation approach offer at least the possibility of constraining a unique solution for age and erosion rate, or does that vanish with this approximation? This could be tested with a carefully composed pseudo-profile that characterizes the muon cross depth. I'd be more convinced of the acceptability of the approximation if the authors could show this. I'm still a bit concerned that there might be an issue here with deep profiles.*

We apologize for the mistake. Whether calculated using erosion rate or eroded thickness, our least-squares approach requires external information (or assumption) for the erosion rate/thickness, therefore cannot be used to calculate a unique solution for age and erosion rate. We also find this sentence may not be suitable for our manuscript, therefore we deleted it from our revision.

We also included an example in section 3.1.4 to show how our approach performs with deep profiles.

*Line 127-130: What about uncertainty in density? I realize that this is basically an uncertainty in depth (assuming the authors are accounting for mass-depth), but it is unclear exactly how uncertainty in mass depth is applied as it can include both a random (individual samples) and systematic component (effective depth shifting of all samples). Also, how does uncertainty in inheritance factor in at this stage?*

We didn't mention uncertainty in density, but we have considered it in our approach. The uncertainty in inheritance is the outcome of the simulation.

In the revision, we have added this to L135-137, L141.

*Line 135: what corresponding probability density functions are used?*

Either uniform or normal distributions. We clarified this in our revision (L141-142).

*Line 137-139: how are the probability density functions calculated from the simulation results? Are the results weighted somehow, or is this a histogram vs. a pdf? It appears to be a histogram.*

We added pdf into our codes and updated the figures in our revision (Fig.8 and 9).

*Table 3: In the Hidy et al. 2010 model of Lees Ferry, muons are not approximated with a two-term exponential (it uses a 5-term approximation like Schaller et al. 2002 and is internally optimized for the sample site and specific depth range). Also, the erosion rate range used was 0-0.4 cm/kyr. These differences should be noted.*

Thank you for pointing this out. We have added a footnote to the table to clarify this difference. We have also included the difference between muon approximation in the discussion section (4.1, L325-327). In our revision, we have deleted the erosion-rate approach part and only use the eroded-thickness approach (Section 3.2).

*Line 216: Where does the 0-0.32 cm/kyr erosion rate estimate come from?*

We find that using the full range of erosion rate (0 to 0.4 cm/kyr) results in an age range that is too old and inconsistent with the total erosion of 0 to 30 cm. This is because the upper bound on total erosion limits the range of acceptable results in the Hidy et al. (2010) model. Because our least-squares approach uses either erosion rate or total eroded thickness, but not both, we chose to set a narrower limit on the erosion rate, from 0-0.32 cm/kyr, so that the ages and total eroded thickness were consistent with that cited in the original paper. Using our eroded thickness fitting approach also avoids this problem.

In our revision, we have deleted the erosion-rate approach part and only use the eroded-thickness approach (Section 3.2).

*Line 261-263: Not allowing negative inheritance actually changes the best-fit, or the peak in the distribution? I see how this would, and philosophically should, change the shape of the full distribution, but it shouldn't have an impact on the best fit—otherwise what makes it best? I guess it might because these are not probability density functions being generated, but histograms. So, doing this might actually be OK in the context of their modeling approach, but I'm hesitant to agree since I am unsure how all those allowable solutions with negative inheritance might introduce artefacts in other solution spaces.*

In the revision, we specifically stated that the negative inheritance effect is important for our approach (L394-395). We also demonstrate this effect with simulated depth profile examples in section 3.1.2.

*Line 272-273: In the originally published Hidy et al (2010) Lees Ferry result, generous uncertainties in 10Be half-life (5%) and muon production (10%; probably still realistic considering Balco 2017) were applied, so it would be useful to know what uncertainties were applied here for comparison. This could also explain some of the differences in results between those histograms and these. Also, out of curiosity, I reran the original Lees Ferry dataset using the Bayesian version of the Hidy et al. (2010) model that generates actual probability density functions vs. the histograms of the original— basically by weighting all MC generated profiles (including solutions outside 95%) by the chi-squared likelihood function. Note that this is very different from what was presented in Hidy et al. (2010), but it is the version that has been adopted since 2012 so probably what should be used for a results comparison. With version 1.2, the results for age at 95% confidence are 76.6 – 96.1 ka (see figure below), with the probability weighting significantly tightening the distribution.*

The version 1.2 we had gave us a resulting age range of 74.5 – 98.7 ka (we used the predefined Lees Ferry settings that came along with the program). We are not sure the source of this discrepancy. We are happy to use the new age range you present here (L295).

We find the difference of the two approaches may come from the following two sources: first, our examples with pseudo depth profiles demonstrated that different approaches always return different estimates; second, we use a two-term exponential for muogenic process instead of a 5-term approximation. We have included this in our revision (L323-327)

*Lines 299-302: Yes, this can't be overstated! Also, there are numerous references out there that support the importance of soil processes to interpreting TCN profiles.*

We included references in our revision (L349-350).

*Lines 310-314: This is an interesting exercise, but are there approaches that ignore radioactive decay? Strange if there is.*

We removed this section from the revision.

*Lines 330-332: Generally, I agree with this, but there are instances where dating highly eroded surfaces are useful when one is more interested in soil age vs. deposition age.*

We agree. In the revision, we used simulated depth profiles (section 3.1.3) and error analysis (4.2.2) to show that our model can still provide accurate enough estimations for highly eroded surfaces.

*Lines 368-370: True, this isn't really a revelation though and is why many depth profiles end up reported with zero-erosion rate minimum ages when constraints on surface erosion can't be justified.*

We agree. In the revision, we have rearranged and deleted this section.

*Lines 399-401: I disagree with this statement. While it may be true for this modeling approach, it is incorrect to infer for all inversion models that may apply different statistical methods for reporting solutions.*

In the revision, we have specifically stated that the negative inheritance effect is important for our approach (L394-395).

---

## Referee Report (RR1)

**Review of Wang and Oskin, 'Combined linear regression...'**
* * *
**Summary.** As noted by some of the reviewers, to some extent this paper is a solution in search of a problem. Cosmogenic-nuclide depth profiles are usually interpreted by inversion of a forward model that predicts nuclide concentrations and whose parameters are the age of a surface, the erosion rate of a surface, and some nuisance parameters typically including inherited nuclide concentrations. Generally this approach works fine, or as well as can be expected given the inherent lack of age resolution for typical depth profiles in which the muon-produced inventory is small in relation to the inheritance.

This paper presents a nice demonstration that you can perform a very simple inversion of depth-profile data for age and inheritance using linear regression, if you change coordinates from depth to production rate. This does help to simplify the problem somewhat and make it more easily accessible, although probably not as much as suggested by the authors. Thus, from this perspective I think this is a contribution that is certainly of interest for publication. However, there are some issues that have come up in review that I do think need further attention, as described below.

**Why needed?** One of the issues that has come up in the discussion of this paper is that from the applications perspective there is not a strong need for a simplified inversion. Depth-profile data are not collected in great quantity and there is not a science application that is currently seriously hindered by the computational time needed to do a full forward model inversion. For myself I am not worried about this issue and I don't think it's an obstacle to publication. For one thing, it is potentially useful for making sure that a more complex inversion scheme is working correctly. Also, I can envision a fast inversion method being useful for database applications in which one seeks to compare a lot of depth-profile results using different production rate scaling methods, or something of that nature. Of course it's actually not that fast because you still need to estimate all the production rates due to muons, which requires a site-specific muon production calculation, which in turn requires a bunch of numerical integrations no matter what. Regardless, however, even though a simplified inversion method isn't of dire immediate need for any current application, it is certainly something that is potentially useful.

**The issue of negative inheritance.** On the other hand, a second issue that has been the focus of much of the review discussion seems to be a more serious problem. This has to do with error propagation and generating an uncertainty distribution for the surface exposure age. It is not really feasible to come up with an analytical expression for the uncertainty in $T_e$ and $C_{inh}$, or even to use something like a York regression, because many of the input uncertainties (e.g., in $P_{zi}$, or in the mass depths because they all depend on the same density measurements) are correlated in a complicated way. Thus, the authors use a Monte Carlo simulation where they vary the input parameters and carry out the regression many times to generate an uncertainty distribution. This brings up the issue, which has been discussed at length in the review comments, of how to handle the fact that many Monte Carlo realizations generate negative values for nuclide inheritance. The authors have proposed, and discuss in both paper drafts and the response to reviews, two methods for dealing with this: first, accepting all Monte Carlo results even if they yield unphysical negative values of the inheritance; second, discarding as unphysical Monte Carlo realizations that yield negative values for the inheritance.

Here I will argue that both of these approaches are incorrect. Although this overall subject seems like a rather arcane point, similar situations often occur in cosmogenic-nuclide applications where a forward model that includes inheritance is being fit to data. Thus, I think airing this issue in the discussion of this paper is valuable and helpful for the field overall.

I think the reason that both approaches are incorrect is that a simple linear regression for $C_{inh}$ and $T_e$ given $C = PT_e + C_{inh}$ (here I am just abbreviating Equation 4a by replacing all the production terms with $P$) is an incomplete description of the regression problem. In other words, even if one leaves aside the question of whether linear regression is the best way to determine the age, the authors have not defined the regression problem itself appropriately.

Basically, linear regression is a least-squares optimization problem:

Problem 1:

$$\text{given} \qquad\qquad\qquad\qquad\qquad X = (T_e, C_{inh}) \qquad\qquad\qquad (1a)$$

$$\text{minimize} \qquad f(X) = \sum_i \left[ (C_{inh} + P_i T_e) - C_i \right]^2 \qquad\qquad (1b)$$

$$\text{over X such that: } -\infty < T_e < \infty; -\infty < C_{inh} < \infty \qquad\qquad (1c)$$

$$(1d)$$

This is how the authors are treating it, as an unconstrained optimization problem in which you can optimize over all $X$, meaning any possible values of $T_e$ and $C_{inh}$. This leads to the normal simple formulae for linear regression.

Unfortunately this is an incomplete description of the problem. Because of the physical constraints that age and inheritance can't be negative, the problem is actually something different:

Problem 2:

$$\text{given} \qquad\qquad\qquad\qquad\qquad X = (T_e, C_{inh}) \qquad\qquad\qquad (2a)$$

$$\text{minimize} \qquad f(X) = \sum_i \left[ (C_{inh} + P_i T_e) - C_i \right]^2 \qquad\qquad (2b)$$

$$\text{over X such that: } \qquad 0 < T_e < \infty; 0 < C_{inh} < \infty \qquad\qquad (2c)$$

$$(2d)$$

Problem 1 and Problem 2 are equivalent only if the constraints are inoperative, which occurs when $C_{inh} \gg 0$ and $T_e \gg 0$. If that's not the case, then the problems are not equivalent. Solving the unconstrained problem won't necessarily also give the answer to the constrained problem.

Unfortunately the constrained problem requires a numerical solution. Non-negative least squares problems are common enough to have a Wikipedia entry, so there are canned algorithms in many programming environments. Regardless, this causes some trouble in the context of the paper because a key point of the paper is that linear regression can be done with a simple formula, not a numerical optimization. That is, you can solve Problem 1 just by writing down the linear regression formula. If you add constraints to the optimization, this isn't true any more – you have to use a numerical optimization scheme. So if you have to consider the complete problem – Problem 2 – you have to use a numerical method anyway and much of the simplicity advantage of linear regression disappears.

Of course, the difference between Problem 1 and Problem 2 mostly doesn't affect inverting the depth profile once by linear regression – the initial attempt would only fail if the initial regression from the data as measured gives $T_e < 0$ or $C_{inh} < 0$, which isn't going to happen in most applications. It only affects figuring out what the uncertainty distribution is in a Monte Carlo simulation. If either the age or the uncertainty are near zero, then solving the unconstrained problem repeatedly and discarding all iterations that yield $C_{inh} < 0$ (or $T_e < 0$, which is not mentioned in the paper but could happen) will NOT give the same result as solving the constrained problem repeatedly.

Consider three possibilities:

- Option 1. Solve Problem 1 in each iteration.

- Option 2. Solve problem 1 in each iteration, but discard all iterations in which $C_{inh} < 0$.

- Option 3. Solve problem 2, the constrained optimization, using a numerical optimizer in each Monte Carlo iteration.

[Figure]

Figure 1: *Monte Carlo uncertainty estimates for a simplified regression problem with an age of 20 ka and inheritance equal to 1 ka of exposure.*

If the constraints are inoperative ($C_{inh} >> 0$ and $T_e >> 0$) these all give the same result. However, consider Figure 1, which shows Monte Carlo inversion results for a simple test problem where the age is 20,000 years and the inheritance is equal to 1000 years exposure, so the constraints are operative. Option 1 (unconstrained, no clipping) results in normal uncertainty distributions for concentration and age, but the uncertainty distribution for inheritance incorrectly extends to negative values. Option 2 removes all iterations that yield negative inheritance, which, obviously, yields a clipped normal distribution for inheritance, but less obviously produces a low-skewed distribution for the age, because the steeper slopes that led to negative intercepts on the inheritance axis are discarded. Option 3, the correct constrained optimization approach, forces all values of the inheritance that would have been negative in the unconstrained problem to be exactly zero, which leads to a secondary mode at the high end of the age distribution. Thus, both Option 1 and Option

2 lead to incorrect estimates of the uncertainty distribution. In this example, Option 1 incorrectly indicates that there is a finite likelihood that the age is greater than ∼22 ka. Option 2 incorrectly underestimates the likelihood that the age is between ∼20 and ∼22 ka.

This is, in fact, rather weird behaviour, which maybe is a hint that linear regression is maybe not the best way to generate uncertainty distributions when age or inheritance is close to zero. It is certainly possible that what we should be learning here is that if we want an uncertainty distribution for the age when inheritance is close to zero, we should use something different, possibly more like a Bayesian fitting scheme with a prior restricted to positive age and inheritance.

So, the authors are correct in remarking in their response that only Option 1 leads to a normal uncertainty distribution. However, their assertions that "negative results, though physically impossible, are necessary for mathematical reasons" and "the overall distributions of the inheritance should be centered around 0, meaning that approximately half of the estimated inheritance should be negative" are, in my opinion, not correct. In fact, the physical requirement is that NONE of the estimated inheritance values (or the age) should be negative, which is why the regression problem is required to be a constrained and not an unconstrained optimization. This is not expected to lead to a normal or symmetrical uncertainty distribution. It implies non-normal uncertainty distributions for both the age and inheritance (which, by the way, should not be represented by means and standard deviations, but rather statistics that aren't specific to a particular distribution, for example mode and confidence intervals).

This effect is more striking (although less pathological-looking) if the inheritance really is close to zero. Figure 2 shows the same results for an age of 10 ka and zero inheritance. Here the correct uncertainty distribution for the age (Option 3) is extremely skewed and of course representing it as a mean and standard deviation would be very misleading.

[Figure]

Figure 2: *Monte Carlo uncertainty estimates for a simplified regression problem with an age of 10 ka and zero inheritance.*

To summarize, the question here isn't really which uncertainty distribution is correct (I would say that

once you have chosen linear regression as the overall method, only option 3 is correct), but whether the uncertainty distributions generated by either Option 1 or Option 2 are close enough to the correct answer that the inaccuracy incurred by solving an oversimplified problem is outweighed by the convenience of using the simple unconstrained regression formula. Basically, the answer to this is that if $C_{inh} >> 0$ and $T_e >> 0$, the uncertainty distribution derived from the unconstrained problem is totally fine. If this isn't true, the unconstrained problem gives the wrong uncertainty distribution, although it might not be that different, so this might not matter in many applications.

In summary, I apologize for discussing at great length the issue of negative inheritance, which is, in fact, an arcane side issue that only affects the uncertainty analysis and is not the main point of this paper. However, as noted, similar issues come up in other cosmogenic-nuclide applications and I think a thorough discussion of the issue is helpful. Regardless, I think the regression model discussed in this paper is interesting and potentially useful, and I am supportive of publication. However, I think the authors do need to revise discussion of the negative inheritance issue. Specifically,

1. As written, several parts of section 3.1.2 are not correct, for example "imposing the physically reasonably prerequisite...may lead to underestimation of the exposure age." It's not a physically reasonable but optional prereequisite, it's a requirement. It's also inaccurate to say that it leads to underestimation of the exposure age – what it actually does is lead to an incorrect uncertainty distribution, and the lower value of the age appears to be the result of improperly using the mean to represent an asymmetrical distribution. In the constrained regression problem, the uncertainty distribution is not expected to be symmetrical about the true value. Thus, this section needs revision so that it correctly outlines how (i) the actual regression problem is a constrained linear regression that is expected to lead to complicated uncertainty distributions when the constraints are operative, and (ii) applying an unconstrained regression is a simplification that is only correct when $C_{inh} >> 0$ and $T_e >> 0$.

2. It's also probably worth a brief discussion of the case where a surface is quite young, so that the uncertainty distribution for t runs into zero. The unconstrained regression is also inappropriate in this case. This could possibly occur even when t is fairly large if the inheritance is also large, such that $T_e P < C_{inh}$.

3. The Beida River / Fig 8 analysis, which improperly uses an unconstrained regression when constraints are operative, should be redone with the correct, constrained regression. Of course for the Lees Ferry analysis, the constraints are inoperative, so the unconstrained regression is fine.

In any case, items 1-3 above are the main changes that I think are needed for publication.

**Other items.** In addition, there are a few lesser items that should be corrected before publication.

The most important one is that the description of the effective attenuation lengths for muon production (line 62) is oversimplified and therefore somewhat misleading. There is no single attenuation length for either fast muon production or negative muon capture production, because the nature of the production process is such that as depth increases, the energy of the remaining muons increases, so the instantaneous attenuation length for the production process also increases. Thus, describing $\Lambda_{m1}$ and $\Lambda_{m2}$ as the attenuation lengths for these processes is not correct. The values of 1500 and 4320 g cm$^{-2}$ that are in Table 1 in this paper, which were given by Heisinger as approximate values that could be used in simplified erosion rate integrations, are not correct at any site or depth, except possibly by accident. Using these values in an application with significant muon production would most likely yield a result that was quite wrong. However, it is true (see Balco, 2019, section 8) that it is usually possible to represent total production by muons in a finite depth range at a specific site as the sum of two exponential functions. Although the authors' response to the reviews stated that "We

updated muon production rates used in the pseudo profiles and in the Beida River case,' this is not evident from the text, and as far as I can tell, it appears likely that what they actually did was modify the surface production rates and not the attenuation constants. More seriously, it is not possible to tell whether they used site-specific values for the Beida River and Lees Ferry example, or if they used the incorrect values in Table 1. At the very least, the authors should modify the text near line 62 to indicate that the values of $P$ and $\Lambda$ pertaining to muon production are not, in fact, true production rates or attenuation lengths for any particular process, but instead are site-specific constants that come from fitting to a more complicated production model.

Two lesser issues just pertain to reducing confusion in the paper:

Starting in section 3 of the paper the authors contrast the results of their regression scheme with a full forward-model-fitting scheme that they describe as 'Bayesian.' This is confusing because the important point is that this method employs a full forward model to predict the observables – the contrast between this method and the regression method would be the same whether the approach to choosing the best values of the model parameters approach was Bayesian, frequentist, or something else. Likewise, it would be possible to perform a Bayesian linear regression. Thus, it would be more helpful to the reader to describe this alternative method as 'forward model fitting' or 'forward model optimization' rather than 'Bayesian' by itself.

Near line 40, the authors use the term 'derivative' to describe refined approaches that are derived from the initial regression approach. The use of this word is confusing here, because in the context of a paper like this one that is mathematical in nature, it implies to the reader that these approaches will involve derivatives in the mathematical sense. For example, in line 44, the use of 'second derivative approach' indicates to the reader that the approach will involve second derivatives of some function or field, which is not the case. Thus, 'derivative' should not be used here. Possible improvements would be to use 'specific approaches' or 'special cases' to contrast with the 'general approach' in line 40.

Finally, a thoroughly insignificant point is that I disagree with reviewer Alan Hidy about the abbreviation 'TCN.' I see no reason that the method in the paper could not be applied to depth profiles in lunar or Martian regolith! The authors should use their best judgement here.

---

## Editor Decision (ED1)

Prof. Pieter Vermeesch
University College London
+44 (0)20 3108 6369
http://ucl.ac.uk/~ucfbpve/

12 July 2022

Dear Dr. Wang and Dr. Oskin,

I apologise once again for the lengthy review process. It has not been easy to find qualified reviewers for your manuscript. However, Dr. Greg Balco kindly stepped in and provided a detailed and constructive review of your paper that largely agrees with my own opinion.

You have done a good job at addressing the comments on the first version of your paper, with one exception. The issue of negative inheritance, which I had raised in my own review of your original manuscript, is brushed aside and not dealt with in earnest.

I agree with Dr. Balco that this issue must be addressed in the final version of your paper, not only because it affects your uncertainty estimates, but also because it has wider implications for cosmogenic nuclide geochronology in general.

The reviewer succinctly summarises three inverse modelling approaches using two compact sets of equations. He argues that the second set of equations ('Problem 2') provides more meaningful uncertainty estimates than the unconstrained approach ('Problem 1') proposed in your paper. Dr. Balco's 'Problem 2' formulation is similar to the suggestion that I made in my review of your first submission:

> "A pragmatic way to fix this issue is to parameterise your problem in terms of the logarithm of inheritence, rather than inheritence itself. After the optimisation, you can then exponentiate the maximum likelihood value to ensure a strictly positive result."

Using similar notation to that used in Dr. Balco's review, the latter approach can be implemented by defining two new variables $\gamma$ and $\tau$ so that $C_{inh} = e^\gamma$ and $T_e = e^\tau$. Then:

$$\text{given } X = (\gamma, \tau) \tag{3a}$$

$$\text{minimise } f(X) = \sum_i \left[(e^\gamma + P_i e^\tau) - C_i\right]^2 \tag{3b}$$

$$\text{over } X \text{ such that } -\infty < \gamma < \infty; -\infty < \tau < \infty. \tag{3c}$$

The resulting uncertainty distributions for $C_{inh}$ and $\tau$ will be skewed (as in Dr. Balco's example), but will (probably) be unimodal rather than bimodal.

**Prof. Pieter Vermeesch**
**University College London**
**+44 (0)20 3108 6369**
`http://ucl.ac.uk/~ucfbpve/`

[Figure]

Regardless of which approach you prefer, it is important that you discuss these options in the next iteration of your manuscript. You should also address the other points raised by Greg Balco. I will then make a final decision without requiring another round of review.

Sincerely yours,

Pieter Vermeesch

---

## Author Response (AR2)

Dear Prof. Vermeesch:

Thank you for your acknowledgement of our effort to revise our first submission, and for the effort in searching for qualified reviewers for our manuscript. We appreciate the comments and suggestions from Dr. Balco and you, and we have made changes to our manuscript accordingly.

We concede to Dr. Balco's and your argument that the inheritance and effective exposure age must be constrained as positive during inversion. Therefore, we have rewritten the related section (3.1.2) along with appropriate simulation examples to emphasize the importance of doing a constrained least squares inversion when either exposure age or inheritance are small and negative results may arise. We have also revised the codes that we uploaded to GitHub to return constrained results.

We also appreciate your suggestion of changing the inheritance and effective exposure age into exponential forms. We have investigated this transition, but we are unable to transform it into a linear inversion problem. We therefore decided to keep the original form in order to preserve the simplicity of the approach.

We also made some minor revision following Dr. Balco's suggestions, as described below:

Reply to comments from Dr. Balco:

*Why needed?* One of the issues that has come up in the discussion of this paper is that from the applications perspective there is not a strong need for a simplified inversion. Depth-profile data are not collected in great quantity and there is not a science application that is currently seriously hindered by the computational time needed to do a full forward model inversion. For myself I am not worried about this issue and I don't think it's an obstacle to publication. For one thing, it is potentially useful for making sure that a more complex inversion scheme is working correctly. Also, I can envision a fast inversion method being useful for database applications in which one seeks to compare a lot of depth-profile results using different production rate scaling methods, or something of that nature. Of course it's actually not that fast because you still need to estimate all the production rates due to muons, which requires a site-specific muon production calculation, which in turn requires a bunch of numerical integrations no matter what. Regardless, however, even though a simplified inversion method isn't of dire immediate need for any current application, it is certainly something that is potentially useful.*

Thank you for your comment. We appreciate for your acknowledgement of the potential of our approach. We envision our approach as a compliment to existing forward models, and it is a convenient tool to explore the trade-offs between exposure age and erosion rate, as we state in the revised introduction: "As an inverse approach, linear regression directly returns a best estimate, providing a clear way to explore how model inputs, such as erosion, affect the result."

*The issue of negative inheritance.*

*1. As written, several parts of section 3.1.2 are not correct, for example "imposing the physically reasonably prerequisite...may lead to underestimation of the exposure age." It's not a physically reasonable but optional prereequisite, it's a requirement. It's also inaccurate to say that it leads to*

*underestimation of the exposure age – what it actually does is lead to an incorrect uncertainty distribution, and the lower value of the age appears to be the result of improperly using the mean to represent an asymmetrical distribution. In the constrained regression problem, the uncertainty distribution is not expected to be symmetrical about the true value. Thus, this section needs revision so that it correctly outlines how (i) the actual regression problem is a constrained linear regression that is expected to lead to complicated uncertainty distributions when the constraints are operative, and (ii) applying an unconstrained regression is a simplification that is only correct when Cinh >> 0 and Te >> 0.*

*2. It's also probably worth a brief discussion of the case where a surface is quite young, so that the uncertainty distribution for t runs into zero. The unconstrained regression is also inappropriate in this case. This could possibly occur even when t is fairly large if the inheritance is also large, such that TeP < Cinh.*

Thank you for the detailed analysis and explanation of the negative inheritance problem. We agree with the reviewer, and we have rewritten section 3.1.2 accordingly. We have also corrected the related statements in section 4.2.3.

*3. The Beida River / Fig 8 analysis, which improperly uses an unconstrained regression when constraints are operative, should be redone with the correct, constrained regression. Of course for the Lees Ferry analysis, the constraints are inoperative, so the unconstrained regression is fine.*

We have updated the data, figure, and analysis for the Beida River, based on the constrained model.

*Other items.*

*The most important one is that the description of the effective attenuation lengths for muon production (line 62) is oversimplified and therefore somewhat misleading. There is no single attenuation length for either fast muon production or negative muon capture production, because the nature of the production process is such that as depth increases, the energy of the remaining muons increases, so the instantaneous attenuation length for the production process also increases. Thus, describing m1 and m2 as the attenuation lengths for these processes is not correct. The values of 1500 and 4320 g cm2 that are in Table 1 in this paper, which were given by Heisinger as approximate values that could be used in simplified erosion rate integrations, are not correct at any site or depth, except possibly by accident. Using these values in an application with significant muon production would most likely yield a result that was quite wrong. However, it is true (see Balco, 2019, section 8) that it is usually possible to represent total production by muons in a finite depth range at a specific site as the sum of two exponential functions. Although the authors' response to the reviews stated that "We updated muon production rates used in the pseudo profiles and in the Beida River case,' this is not evident from the text, and as far as I can tell, it appears likely that what they actually did was modify the surface production rates and not the attenuation constants. More seriously, it is not possible to tell whether they used site-specific values for the Beida River and Lees Ferry example, or if they used the incorrect values in Table 1. At the very least, the authors should modify the text near line 62 to indicate that the values of P and Λ pertaining to muon production are not, in fact, true production rates or attenuation lengths for any*

*particular process, but instead are site-specific constants that come from fitting to a more complicated production model.*

Thank you for pointing this out. We realize that the statement we made in the manuscript is misleading, therefore, we modified the text originally in L62 as "Note that the two exponential terms for muogenic production used here are an approximation of a complex muogenic production path, because the muon energy spectrum hardens continuously with depth (Helsinger 2002a, 2002b; Marrero et al., 2016; Balco 2017)." (L64-66 in the revision), we also added a footnote to Table 1 and S2 to clarify this. For both the Beida River and the Lees Ferry examples, we updated the inversion results using site specific muon production rate and attenuation length based on Model 1B of Balco, 2017. This leads to less than 1% of difference between the old and revised age estimations (Section 3.2).

*Starting in section 3 of the paper the authors contrast the results of their regression scheme with a full forward-model-fitting scheme that they describe as 'Bayesian.' This is confusing because the important point is that this method employs a full forward model to predict the observables – the contrast between this method and the regression method would be the same whether the approach to choosing the best values of the model parameters approach was Bayesian, frequentist, or something else. Likewise, it would be possible to perform a Bayesian linear regression. Thus, it would be more helpful to the reader to describe this alternative method as 'forward model fitting' or 'forward model optimization' rather than 'Bayesian' by itself.*

Thank you for pointing this out. We agree with the reviewer that using Bayesian in our previous revision is confusing for the readers. We have replaced 'Bayesian' with 'forward model fitting' and 'forward approach' in the revision.

*Near line 40, the authors use the term 'derivative' to describe refined approaches that are derived from the initial regression approach. The use of this word is confusing here, because in the context of a paper like this one that is mathematical in nature, it implies to the reader that these approaches will involve derivatives in the mathematical sense. For example, in line 44, the use of 'second derivative approach' indicates to the reader that the approach will involve second derivatives of some function or field, which is not the case.*

*Thus, 'derivative' should not be used here. Possible improvements would be to use 'specific approaches' or 'special cases' to contrast with the 'general approach' in line 40.*

Thank you for the suggestion, we have changed the 'derivative' approaches into 'specific approaches'.

*Finally, a thoroughly insignificant point is that I disagree with reviewer Alan Hidy about the abbreviation 'TCN.' I see no reason that the method in the paper could not be applied to depth profiles in lunar or Martian regolith! The authors should use their best judgement here.*

We have replaced 'TCN' with 'CN'.

---

## Author Response (AR3)

Dear Prof. Vermeesch,

Thank you for your suggestion! We agree that we should mention the effect of negative inheritance/age upfront. Therefore, we have made the following changes to our manuscript:

For the abstract, we changed the first sentence as "…, while retaining the advantages of linear inversion for surfaces with inheritance and age much greater than zero." We also added a sentence in line 13-14: "For surfaces with very low inheritance or age, it is important to apply a constrained inversion to obtain the correct result distributions."

For the "Conclusion" section, we added "For surfaces with low inheritance or age, as demonstrated with the Beida River sample site and the simulated profiles, unconstrained inversion could lead to incorrect distributions for inversion results. Therefore, the linear regression we proposed here requires applying the nonnegative constraint on such depth profile data." in line 410-412.

We appreciate the time and effort you and reviewers put into our manuscript. We hope our revision meets your expectations.

Yiran Wang

Track-changes for the supplement (previous revision):

**Combined linear regression and Monte Carlo approach to modelling exposure age depth profiles**

**Supporting information**

Yiran Wang[1, 2], Michael E. Oskin[1]

[1]Department of Earth and Planetary Sciences, UC Davis, Davis, 95616, USA
[2]Earth Observatory of Singapore, Nanyang Technological University, 639798, Singapore

*Correspondence to*: Yiran Wang (yrwwang@ucdavis.edu)

**1. Simulated CN depth profiles**

Our simulation process is conducted with following steps:

1. For each scenario, we first produce an ideal depth profile based on given exposure age, inheritance, production rate, attenuation, density, etc.

2. Next, we produce a suite of simulated depth profiles with each sample deviated from the true value based on an imposed error distribution which defined as "deviation of sample concentration. In addition, an independent analytical uncertainty is also assigned for each sample. The mean of the sample depth is the same as the true value; but an uncertainty is assigned for each sample depth. The remaining parameters (density, production rate, attenuation) are the same as the ideal profile, with no imposed uncertainty.

3. For each profile within the suite, we estimate the exposure age and inheritance based on the samples generated in step 2, using both least-squares linear regression and forward modeling (Bayesian Monte Carlo) approaches.

4. From the analysis suite we compare the resulting distributions of the estimated age and inheritance using both methods with the true value and with each other.

**1.1 Deviation of sample concentrations**

[Figure]

[Figure]

[Figure]

[Figure]

**Figure S1 Distributions of age (a and b) and inheritance (c and d) estimation results for 500 simulated CN profiles with 2% imposed deviation of sample concentration. a. Distribution of exposure age, sorted by mean age estimated from linear regression (eq. 4). b. Distribution of exposure age, sorted mean age estimated using a forward approach. c. Distribution of inheritance, sorted by mean inheritance estimated from linear regression (eq. 4). d. Distribution of inheritance, sorted by mean inheritance estimated using a forward approach.**

[Figure]

[Figure]

**Figure S2 Distribution of age (a, b, e, f) and inheritance (c, d, g, h) estimation results for 500 simulated CN profiles with 5% imposed deviation of sample concentration. a. and b. Histogram of the mean exposure age estimated from linear**

regression and a forward approach. c. and d. Histogram of mean inheritance estimated from linear regression a forward approach. e. Distribution of exposure age, sorted by mean age estimated from linear regression (eq. 4). f. Distribution of exposure age, sorted mean age estimated using a forward approach. g. Distribution of inheritance, sorted by mean inheritance estimated from linear regression (eq. 4). h. Distribution of inheritance, sorted by mean inheritance estimated using a forward approach.

[Figure]

[Figure]

**Figure S3 Distribution of age (a, b, e, f) and inheritance (c, d, g, h) estimation results for 500 simulated CN profiles with 10% imposed deviation of sample concentration. a. and b. Histogram of the mean exposure age estimated from linear**

regression and a forward approach. c. and d. Histogram of mean inheritance estimated from linear regression a forward approach. e. Distribution of exposure age, sorted by mean age estimated from linear regression (eq. 4). f. Distribution of exposure age, sorted mean age estimated using a forward approach. g. Distribution of inheritance, sorted by mean inheritance estimated from linear regression (eq. 4). h. Distribution of inheritance, sorted by mean inheritance estimated using a forward approach.

**1.2 Denudation depth**

[Figure]

~~**Figure S4 Distribution of estimation results for 500 simulated zero inheritance TCN profiles with 5% imposed deviation of sample concentration. a-e. Histogram of the mean inheritance estimated from linear regression and a Bayesian approach under different inversion schemes: not permitting negative inheritance during inversion (a and d), permit negative inheritance during inversion (b and e), excluding negative inheritance results after inversion (c, linear inversion only). f. Distribution of exposure age estimated from linear regression, sorted from mean age estimated by not permitting negative inheritance during inversion. g. Distribution of exposure age estimated from linear regression, sorted from mean age estimated by excluding negative inheritance after inversion. h. Distribution of exposure age estimated from a Bayesian approach, sorted from mean age estimated by not permitting negative inheritance during inversion.**~~

[Figure]

Figure S5 Distribution of estimation results for 500 simulated low inheritance (5000 atoms/g) TCN profiles with 5% imposed deviation of sample concentration. a-e. Histogram of the mean exposure age estimated from linear regression and a Bayesian approach under different inversion schemes: not permitting negative inheritance during inversion (a and d), permit negative inheritance during inversion (b and e), excluding negative inheritance results after inversion (c, linear inversion only). f-j. Histogram of the mean inheritance estimated from linear regression and a Bayesian approach under different inversion schemes: not permitting negative inheritance during inversion (f and i), permit negative inheritance during inversion (g and j), excluding negative inheritance results after inversion (h, linear inversion only). k. Distribution of exposure age estimated from linear regression, sorted from mean age estimated by not permitting negative inheritance during inversion. l. Distribution of exposure age estimated from linear regression, sorted from mean age estimated by excluding negative inheritance after inversion. m. Distribution of exposure age estimated from a Bayesian approach, sorted from mean age estimated by not permitting negative inheritance during inversion.

[Figure]

[Figure]

**Figure S4 Distribution of mean exposure age (a-d) and inheritance (e-h) estimated from a forward approach for 500 simulated (5000 atoms/g) CN** profiles with 5% imposed deviation of sample concentration and with total denudation equals to 1 (a and e), 2 (b and f), 3 (c and g) and 5-times (d and h) attenuation length of spallation. Red vertical line annotates the true age and true inheritance.

**1.3 Deep sample profile**

[Figure]

[Figure]

[Figure]

**Figure** S5 **Distribution of estimation results from linear regression and a** forward **approach for 500 simulated** CN **deep (3-5 m) profiles with denudations equal to 0 (a, d, g), 2 (b, e, h), and 5-times (c, f, i) attenuation length, and with 5% imposed deviation of sample concentration. 500 groups of inversion results. a-f. Histograms of the mean inheritance estimated from linear regression (a-c) and a** forward **approach (d-f). g-h. Distribution of exposure age, sorted by mean age estimated from linear regression.**

[Figure]

[Figure]

[Figure]

**Figure** S6 **Distribution of estimation results from linear regression and a** forward **approach for 500 simulated** CN **deep (3-5 m) profiles with denudations equal to 0 (a, d, g), 2 (b, e, h), and 5-times (c, f, i) attenuation length, and with 1% imposed deviation of sample concentration. 500 groups of inversion results. a-f. Histograms of the mean inheritance estimated from linear regression (a-c) and a** forward **approach (d-f). g-h. Distribution of exposure age, sorted by mean age estimated from linear regression.**

**2. Case Examples**

**Table S1 $^{10}$Be concentration and the production rate at each sample depth for the two sample sites.**

| Beida River Terrace (Wang et al., 2020) | | | Lees Ferry Terrace (Hidy et al, 2010) | | |
|---|---|---|---|---|---|
| Sample ID | $^{10}$Be Concentration; $C_1^1$ ($10^5$ atoms/g) | $P_{zn}$ (atoms/(g×yr)) | Sample ID | $^{10}$Be Concentration ($10^5$ atoms/g) | $P_{zn}$ (atoms/(g×yr)) |
| BT2-2-20 | 14.33 ± 0.39 | 13.82 ± 0.91 | GC-04-LF-404.30s | 5.69±0.17 | 6.35 ± 0.48 |
| BT2-2-45 | 9.84 ± 0.36 | 9.94 ± 0.65 | GC-04-LF-404.60s | 4.07±0.11 | 4.09 ± 0.48 |
| BT2-2-75 | 5.68 ± 0.23 | 6.69 ± 0.44 | GC-04-LF-404.100s | 2.92±0.09 | 2.27 ± 0.39 |
| BT2-2-110 | 4.09 ± 0.21 | 4.22 ± 0.28 | GC-04-LF-404.140s | 2.03±0.06 | 1.26 ± 0.29 |
| BT2-2-150 | 2.96 ± 0.11 | 2.84 ± 0.19 | GC-04-LF-404.180s | 1.57±0.05 | 0.7 ± 0.2 |
| BT2-2-180 | 2.63 ± 0.08 | 1.68 ± 0.11 | GC-04-LF-404.220s | 1.34±0.04 | 0.39 ± 0.13 |

1 C1 is the concentration prior to the onset of loess accumulation, following the approach introduced by Hetzel et al., (2004).

**Table S2 Values for parameters used in exposure age calculation.**

| Parameter | Values (Wang et al., 2020) | Values (Hidy et a., 2010) |
|---|---|---|
| Surface production rate (nucleon-negative muon-fast muon) (atom/(g×yr)) | 23.4, 0.0958, 0.0413 [1] | 9.51, 0. 0596, 0.0314 [2] |
| Density (g/cm$^3$) | 2.2 | 2.2-2.5 (uniform distribution) |
| Attenuation (nucleon- negative muon-fast muon) (g/cm$^2$) | 167, 873, 2125 [1] | 160±5, 1070, 2434 [2] |
| Eroded thickness (cm) | 40±10 (normal distribution) | 0-30 (uniform distribution) |

1 The production rate for nucleon is calculated based on the "LSD" scaling scheme (Lifton et al., 2014), the production rates and attenuation length for negative and fast muons are approximated from the site-specific muon production rate at depth using model 1B from Balco, 2017.

2 5-term approximation for muogenic production is applied in the original paper, here we use a 2-term exponential approximation calculated using model 1B from Balco, 2017.